



# Particulate barium tracing significant mesopelagic carbon remineralisation in the North Atlantic

**Nolwenn Lemaitre [1, 2], Hélène Planquette[1], Frédéric Planchon[1], Géraldine Sarthou[1], Stéphanie Jacquet[3], Maribel I. García-Ibáñez[4], Arthur Gourain[1, 5], Marie Cheize[1], Laurence Monin[6], Luc André[6], Priya Laha[7], Herman Terryn[7] and Frank Dehairs[2]**

[1]Laboratoire des Sciences de l'Environnement Marin (LEMAR), UMR 6539, IUEM, Technopôle Brest Iroise, 29280 Plouzané, France

[2]Vrije Universiteit Brussel, Analytical, Environmental and Geo-Chemistry, Earth System Sciences research group, Brussels, Belgium

[3]Aix Marseille Université, CNRS/INSU, Université de Toulon, IRD, Mediterranean Institute of Oceanography (MIO), UM 110, 13288 Marseille, France

[4]Instituto de Investigaciones Marinas, IIM-CSIC, Eduardo Cabello 6, 36208 Vigo, Spain

[5]Ocean Sciences Department, School of Environmental Sciences, University of Liverpool, Liverpool L69 3GP, United Kingdom

[6]Earth Sciences Department, Royal Museum for Central Africa, Leuvensesteenweg 13, Tervuren, 3080, Belgium

[7]Vrije Universiteit Brussel, SURF research group, department of Materials and Chemistry, Brussels, Belgium

*Correspondance to:* Nolwenn Lemaitre, Department of Earth Sciences, Institute of Geochemistry and Petrology, ETH-Zürich, Zürich, Switzerland. (nolwenn.lemaitre@erdw.ethz.ch)

**Keywords:** Particulate biogenic barium; Carbon remineralisation; North Atlantic; Biological pump

**Abstract**

The remineralisation of sinking particles by prokaryotic heterotrophic activities is important for controlling oceanic carbon sequestration. Here, we report mesopelagic particulate organic carbon (POC) remineralisation fluxes in the North Atlantic along the GEOTRACES-GA01 section (GEOVIDE cruise; May-June 2014) using the particulate biogenic barium (excess barium; $Ba_{xs}$) proxy. Important mesopelagic (100–1000 m) $Ba_{xs}$ differences were observed along the transect depending on the intensity of past blooms, the phytoplankton community structure and the physical forcing, including downwelling. The subpolar province was characterized by the highest mesopelagic $Ba_{xs}$ content (up to 727 pmol L$^{-1}$), which was attributed to an intense bloom averaging 6 mg Chl-$a$ m$^{-3}$ between January and June 2014 and by an intense 1500 m-deep convection in the central Labrador Sea during the winter preceding the sampling. This downwelling could have promoted a deepening of the prokaryotic heterotrophic activity, increasing the $Ba_{xs}$ content. In comparison, the temperate province, characterized by the lowest $Ba_{xs}$ content (391 pmol L$^{-1}$), was sampled during



the bloom period and phytoplankton appear to be dominated by small and calcifying species, such as coccolithophorids.
The $Ba_{xs}$ content, related to an oxygen consumption, was converted into a remineralisation flux using an updated
relationship, proposed for the first time in the North Atlantic. The estimated fluxes were in the same order of magnitude
than other fluxes obtained by independent methods (moored sediment traps, incubations) in the North Atlantic.
Interestingly, in the subpolar and subtropical provinces, mesopelagic POC remineralisation fluxes (up to 13 and 4.6
mmol C $m^{-2}$ $d^{-1}$, respectively) were equalling and occasionally even exceeding upper ocean POC export fluxes,
highlighting the important impact of the mesopelagic remineralisation on the biological carbon pump with a near-zero,
deep (> 1000 m) carbon sequestration efficiency in spring 2014.

## 1.   Introduction

The oceanic biological carbon pump (BCP) controls the export of carbon and nutrients to the deep ocean, especially
through the production of biogenic sinking particles. In the North Atlantic, the oceanic BCP is particularly efficient in
transporting carbon to the deep ocean (Buesseler et al., 1992; Buesseler and Boyd, 2009; Herndl and Reinthaler, 2013;
Honjo and Manganini, 1993; Le Moigne et al., 2013b) due to its strong spring bloom (Henson et al., 2009; Pommier
et al., 2009), and it is estimated to contribute up to 18% of the BCP in the world's ocean (Sanders et al., 2014).
However, the efficiency of this transfer depends on many processes including the remineralisation intensity occurring
within the mesopelagic layer (100–1000 m depth layer). In the mesopelagic layer, most of the particulate organic
carbon (POC) exported from surface is released to the dissolved phase, i.e., dissolved organic carbon (DOC; Buesseler
et al., 2007; Buesseler and Boyd, 2009; Burd et al., 2016; Herndl and Reinthaler, 2013; Lampitt and Antia, 1997;
Martin et al., 1987). Mesopelagic remineralisation has been often reported to exceed the carbon supplies (i.e. POC and
DOC; Aristegui et al., 2009; Baltar et al., 2009; Burd et al., 2010; Collins et al., 2015; Fernández-castro et al., 2016;
Giering et al., 2014; Reinthaler et al., 2006), highlighting the importance of the mesopelagic layer on the fate of the
sinking POC, and occasionally questioning the efficiency of the North Atlantic to transfer POC to the deep ocean.
In this context, we examined mesopelagic POC remineralisation along the GEOTRACES-GA01 section during the
GEOVIDE cruise (15 May–30 June, 2014; R/V Pourquoi Pas?) by assessing particulate biogenic barium (excess
barium; $Ba_{xs}$) contents. The accumulation of $Ba_{xs}$ in the mesopelagic layer has been related to the presence of barite
($BaSO_4$) crystals, which appear to be formed by biological activity in the upper water column (Bishop, 1988; Cao et
al., 2016; Dehairs et al., 1980; Horner et al., 2015). The micro-sized barite crystals precipitate inside oversaturated
micro-environments, mostly aggregates of organic material where prokaryotic activity is intense (Bishop, 1988; Collier
and Edmond, 1984; Dehairs et al., 1980; Ganeshram et al., 2003; Gonzalez-Munoz et al., 2003). When these aggregates
are remineralised, barite crystals are released within the mesopelagic layer. Therefore, the $Ba_{xs}$ content can be related
to an oxygen consumption (Dehairs et al., 1997; Shopova et al., 1995) and then converted to a remineralised POC flux
in the mesopelagic layer (Dehairs et al., 1997). $Ba_{xs}$ has been successfully used as a proxy of remineralised POC flux
in the Southern (Cardinal et al., 2005; Jacquet et al., 2008a, 2008b, 2011, 2015; Planchon et al., 2013) and Pacific
Oceans (Dehairs et al., 2008).
This study is the first one to report the use of the $Ba_{xs}$ proxy in the North Atlantic. Regional variations of the $Ba_{xs}$
distributions along the biogeochemical provinces are discussed regarding the stage and intensity of the bloom, the
phytoplankton community structure and the physical forcing. We re-assessed the algorithm between $Ba_{xs}$ content and



oxygen consumption developed for the Southern Ocean, adapting it for the North Atlantic. We compared the
remineralisation fluxes resulting from this new North Atlantic-specific algorithm with those obtained by other methods
in the same area. This comparison, in combination with surface primary production (PP) and POC export estimates
(Lemaitre et al., in prep.), allowed us to investigate the fate of POC to the deep ocean in order to better constrain the
BCP and its efficiency in the North Atlantic.
**2. Methods**
**2.1. Study area**
The GEOVIDE section (15 May–30 June 2014; R/V Pourquoi pas?) crossed different biogeochemical provinces in the
North Atlantic including the North Atlantic subtropical gyre (NAST; Stations 1 and 13), the North Atlantic drift
(NADR) covering the West European (Stations 21 and 26) and Icelandic (Stations 32 and 38) basins, and the Atlantic
Arctic (ARCT) divided between the Irminger (Stations 44 and 51) and Labrador (Stations 64, 69 and 77) Seas
(Longhurst, 1995; Fig. 1, 2).
The evolution of chlorophyll-*a* (Chl-*a*) concentrations from satellite imagery (Fig. 1) revealed the decline of the bloom
in the NAST and the Labrador Sea and the bloom period within the NADR province and the Irminger Sea (Lemaitre
et al., in prep.). Indeed, the highest daily PP rates were measured in the NADR and in the Irminger Sea (> 150 mmol
C m$^{-2}$ d$^{-1}$; A. Roukaerts, D. Fonseca Batista and F. Deman, unpublished data). The phytoplankton community structure
also varied regionally, with diatoms dominating the ARCT province and the West European basin of the NADR,
coccolithophorids dominating the Icelandic basin of the NADR and cyanobacteria in the NAST province (Tonnard et
al., 2017, this issue). Finally, as specifically described elsewhere (Daniault et al., 2016; García-Ibáñez et al., 2015;
Kieke and Yashayaev, 2015; Zunino et al., 2017; this issue), these provinces also differ in terms of their hydrographic
features, in particular, the presence of the sub-arctic front (SAF; which during GEOVIDE was located near Station
26), strong currents near the Greenland margin (probably influencing Stations 51 and 64), and an intense 1500 m-deep
convection in the central Labrador Sea (Station 69) due to the formation of the Labrador Sea Water (LSW) in winter.
These features influenced the magnitude of the carbon export fluxes, as well as the export and transfer efficiencies
along the transect (Lemaitre et al., in prep.). The highest POC export fluxes from the upper-ocean was observed in the
NADR province and in the Labrador Sea, reaching 8.4 ± 0.5 and 10 ± 0.8 mmol C m$^{-2}$ d$^{-1}$ at Stations 32 and 69,
respectively. Export efficiency (e.g. POC export flux divided by PP) was generally low (≤ 12%), except at Stations 1
and 69 where it reached 30%. The transfer efficiency (e.g. upper ocean POC export flux divided by POC export flux
at 400 m) was more variable, ranging from 2% at Station 21 to 78% at Station 64 (Lemaitre et al., in prep.).
**2.2. Sampling and analyses**
In this study, we present two datasets of Ba$_{xs}$ concentrations:
1) Ba$_{xs}$ concentrations measured in samples collected using a standard CTD rosette equipped with 12 L Niskin
bottles. Generally, 18 depths were sampled between surface and 1500 m in order to obtain a high depth resolution of
the Ba$_{xs}$ signal at stations where primary production and POC export fluxes were also determined (Table S1).
Four to eight litters of seawater were filtered on acid-cleaned polycarbonate membranes of 0.4 µm porosity
(Nuclepore®, 47 or 90 mm diameter). Filter membranes were rinsed with Milli-Q water (18.2 MΩ cm; ≤ 5 mL) to





remove sea-salt, dried at ambient temperature under a laminar flow hood and finally stored in clean petri slides until
analysis.
In the laboratory, filters were totally digested with a concentrated tri-acid mixture (1.5 mL HCl / 1 mL HNO$_3$ / 0.5 mL
HF; all Merck suprapur grades) in clean Teflon vials (Savillex®) on a hot plate at 90°C, overnight. Then, the acid
solution was evaporated at 110°C until near dryness, and the residue was dissolved in 13 mL 0.32M HNO$_3$ (Merck;
distilled Normapur). The solutions, in polypropylene tubes (VWR), were analysed for Barium (Ba), Aluminium (Al)
and other major and minor elements using an inductively coupled plasma-quadrupole mass spectrometer (ICP-QMS;
X Series 2 Thermo Fisher) equipped with a collision cell technology (CCT). We used a concentric quartz nebulizer (1
mL min$^{-1}$) and nickel sample and skimmer cones. During the analyses, internal standards (Ru, In, Re and Bi) were
added to samples in order to monitor and correct the instrumental drift and matrix-dependent sensitivity variations.
Two multi-element artificial standard solutions were prepared for external calibration. The first contained major
elements (Na, Mg, Al, Ca and Ti) and the second was prepared with minor elements (Sr, Ba, REEs, Th and U).
Standards were prepared by dilution of the multi-element mixed standard stock solutions to span the expected range
of sample concentrations, with concentrations in the standard curve spaced to cover potential sample variations.
The accuracy and precision of our analyses were assessed using the following Certified Reference Materials (CRM):
BHVO-1, JB-3, JGb-1 and SLRS-5 (Table 1).
The detailed procedure for sample preparation and analysis is given in Cardinal et al. (2001).
2) Ba$_{xs}$ concentrations measured in samples collected using the trace metal clean rosette, equipped with
twenty-two 12 L Go-Flo bottles at higher spatial resolution but with a lower vertical resolution in the mesopelagic
layer. Indeed, the entire water column was sampled at 31 stations along the GEOVIDE transect. Details regarding
filtration, sample processing and analyses can be found in Gourain et al. (2017; this issue). Briefly, at each depth, two
size fractions were investigated: 0.45–5 µm using polysulfone filters (Supor®) and > 5 µm using mixed ester cellulose
filters (MF, Millipore®). Between 2 and 5 L of seawater were filtered for the upper water column (surface to 100 m)
and 10 L below 100 m. In the laboratory, filters were digested with a solution of 8 M HNO$_3$ (Ultrapur grade, Merck)
and 2.3 M HF (Suprapur grade, Merck). Vials were then refluxed at 130 °C on a hotplate during 4 h. After a gentle
evaporation, the residue was brought back into solution with approximately 2 mL of 0.32 M HNO$_3$ spiked with 1 µg
L$^{-1}$ of Indium. Solutions were analysed using a SF-ICP-MS (Element 2, Thermo) following the method of Planquette
and Sherrell (2012). Total Ba and Al concentrations were calculated by adding the two size fractions. The accuracy
and precision of these analyses were assessed using the BCR-414 CRM (see Gourain et al., 2017; this issue).
For both datasets, the Ba$_{xs}$ concentrations were calculated by subtracting the particulate lithogenic barium (pBa-litho)
from the total particulate barium (pBa). The pBa-litho was determined by multiplying the particulate aluminium (pAl)
concentration by the upper continental crust (UCC) molar Ba:Al ratio (0.00135 mol mol$^{-1}$; Taylor and Mclennan,
1985). Along the GEOVIDE section, pBa-litho represented less than 7 % of the total barium, except at Stations 1 and
53, close to the Iberian and Greenland margin, respectively, where pBa-litho accounted for 28 and 44 %, respectively.
Because of the rather large uncertainty associated with the UCC Ba:Al ratio and because of the strong lithogenic
particle loads there, Stations 1 and 53 were not considered further in this study. Uncertainties on Ba$_{xs}$ were estimated
using error propagation and represented from 6 to 25 % of the Ba$_{xs}$ concentrations.



At stations where total pBa and pAl concentrations were available at a similar depth, the comparison between the $Ba_{xs}$
concentrations obtained by the two sampling and analytical methods was excellent (regression slope: 0.94; r²: 71 %;
p<0.01; n=91; Fig. S1).
In addition, analyses of filtered suspended matter were carried out using a Field Emission Scanning Electron
Microscope (FE-SEM; JEOL JSM-7100F) to verify the relationship between $Ba_{xs}$ and barite particles (Section 4.3.1).
For seven samples, we analysed a filter surface of 0.5 cm²: 6 samples with high mesopelagic $Ba_{xs}$ concentrations
(Station 13 at 400 m; Station 38 at 300 m; Station 44 at 300 and 700 m; Station 69 at 600 m and Station 77 at 300 m)
and one sample with high surface $Ba_{xs}$ concentrations (Station 26, 50 m).
**2.3. Determination of Carbon remineralisation fluxes**
In previous studies focusing on the Southern Ocean, $Ba_{xs}$ based-mesopelagic carbon remineralisation fluxes were
estimated using Eq. (1), which relates the accumulated mesopelagic $Ba_{xs}$ inventory to the rate of oxygen consumption
(Shopova et al., 1995; Dehairs et al., 1997):
$$JO_2 = (\text{mesopelagic } Ba_{xs} - Ba_{\text{ residual}}) / 17200 \qquad (1)$$
where $JO_2$ is the rate of oxygen consumption (in µmol L$^{-1}$ d$^{-1}$), *mesopelagic $Ba_{xs}$* is the depth-weighted average in the
mesopelagic layer (DWA; in pmol L$^{-1}$), $Ba_{residual}$ is the deep-ocean $Ba_{xs}$ value observed at zero oxygen consumption
(or $Ba_{xs}$ background signal), which was determined to reach 180 pmol L$^{-1}$ (Dehairs et al., 1997). Then, the oxygen
consumption $JO_2$ can be converted into C remineralised trough Eq. (2):
$$\text{POC mesopelagic remineralisation} = Z \times JO_2 \times (C:O_2)_{\text{Redfield Ratio}} \qquad (2)$$
where the POC mesopelagic remineralisation is in mmol C m$^{-2}$ d$^{-1}$, $Z$ is the thickness of the layer in which the
mesopelagic $Ba_{xs}$ is calculated, $JO_2$ is the rate of oxygen consumption given by Eq. (1) and $(C:O_2)_{Redfield\ Ratio}$ is the
stoichiometric molar C to $O_2$ ratio (127/175; Broecker et al., 1985).
**3. Results**
**3.1. Barite is the main carrier of $Ba_{xs}$**
Several barite particles were observed at proximity to biogenic fragments (Fig. 3) suggesting the important role of the
biogenic microenvironments in barite formation (Bishop, 1988; Dehairs et al., 1980; Stroobants et al., 1991). However,
no barite crystals were observed in surface waters of Station 26 indicating different processes generating high $Ba_{xs}$
concentrations between surface and mesopelagic samples. Indeed, the very high $Ba_{xs}$ concentration in surface waters
of Station 26 (Fig. 5) was not related to barite particles but more likely to Ba uptake and adsorption by biota, as reported
by Sternberg et al. (2005) in culture experiments. This result fits in the concept of barite formation proposed by
Stroobants et al. (1991) showing that the barium sulphate in biogenic aggregates of surface waters is not crystallized
whereas below this surface layer, when degradation occurs, the barite is present as micro-particles. In the mesopelagic
layer, these micro-sized barite particles are characterized by different shapes and sizes (Fig. 3; Dehairs et al., 1980; S.
Jacquet, personal communication).



We evaluated the contribution of the barite particles to $Ba_{xs}$ for the sample collected at 600 m of Station 69 (0.003%
of the total filter surface was analysed), which was selected because of its high mesopelagic $Ba_{xs}$ content (Fig. 5).
Using the Field Emission Scanning Electron Microscope (FE-SEM), we detected the barite particles present in this
surface area and determined their volume. To this aim, each barite particle was pictured using a magnification setting
between 12,000 and 15,000×. Images were then analysed with the software ImageJ and, for each barite particle, the
longest and shortest lengths were measured and converted from pixel to nanometres. Barite particles were then
assimilated to ellipses to deduce their volume. Finally, the concentration of pBa in barite particles was calculated using
Eq. (3):

$$\text{pBa in barite} = \Sigma \left[ V \times \mu_{BaSO4} \times (M_{Ba} / M_{BaSO4}) \right] / V_{SW} \quad (3)$$

where $V$ is the volume of each $BaSO_4$ particle (between 0.01 and 3.96 µm³), $\mu_{BaSO4}$ is the density of barite (4.45 g cm⁻
³), $M_{Ba}/M_{BaSO4}$ is the molar proportion of barium in $BaSO_4$ (0.59) and $V_{SW}$ is the volume of filtered seawater (equivalent
to 0.2 mL through this portion of filter). Assuming that this filter portion is representative of the whole filter, the
concentration deduced from the FE-SEM reached 1260 pmol L⁻¹. This is in the same order of magnitude that the
concentration of total $Ba_{xs}$ analysed by ICP-MS (831 pmol L⁻¹) on the whole filter of this sample. The similarity
between both estimations is remarkable considering the limitations of the FE-SEM procedure (i.e. the very small
fraction of filter analysed). This also confirms that $Ba_{xs}$ is carried by barite particles in the mesopelagic layer, as
observed earlier (Dehairs et al., 1980).

### 3.2. Particulate biogenic $Ba_{xs}$ profiles

#### 3.2.1. Section overview

The high resolution section of $Ba_{xs}$ concentrations (Fig. 4) shows elevated concentrations in the mesopelagic layer
across the section ($Ba_{xs}$ (depth between 100 and 1000 m) = 333 ± 224 pmol L⁻¹; median ± 1s.d.; n=209). In comparison,
the surface ocean (depth < 100 m) and the deep ocean (depth > 1000 m) are characterized by lower median values (94
and 114 pmol L⁻¹, n=113 and 199, respectively). Exceptions can be observed in the upper waters at Stations 25 and 26
and close to the seafloor at Stations 29, 32, 36, 38 and 71, and may be attributed to Ba assimilation by phytoplankton
(Stations 25 and 26), and sediment resuspension (Stations 29, 32, 36, 38, 71; Gourain et al., 2017; this issue).
Concentrations ranged from 4 (Station 11, 55 m) to 24643 (Station 26, 35 m) pmol L⁻¹ in surface waters and from 7
(Station 71, 350 m) to 1388 (Station 15, 300 m) pmol L⁻¹ in the mesopelagic layer (100–1000 m). In the mesopelagic
layer, where the maximum $Ba_{xs}$ concentrations were generally observed, the highest $Ba_{xs}$ concentration was determined
in the NAST province, reaching 1388 pmol L⁻¹ at 300 m of Station 15. However, these maxima occurred in a relatively
narrow depth interval (a layer of 100–300 m), while the maximum were spread over larger depth intervals at other
stations, in particular in the ARCT province (layer of the $Ba_{xs}$ maxima: 1200 m at Station 69).

#### 3.2.2. Individual Profiles

In the following section and if no specific notification are given, only the $Ba_{xs}$ concentrations determined using Niskin
bottles are described because of (1) the good comparison between both Niskin and Go-Flo dataset (regression slope:
0.94; r²: 71 %; p<0.01; n=91); (2) the better resolution in the 100–1000 m layer for Niskin bottles; and (3) the same





cast between Niskin data and 234-Thorium data used for the determination of POC export. A comparison between
primary production, POC export and POC remineralisation is addressed in Section 4.4.
The vertical profiles of $Ba_{xs}$ concentrations at all stations are shown in Figure 5.
In the NAST province (Station 13), the $Ba_{xs}$ concentrations steadily increased from the surface to 400 m, reaching 961
pmol $L^{-1}$, then decreased with depth, reaching the background level of 180 pmol $L^{-1}$ at 1500 m.
In the West European basin of the NADR province, vertical profiles of $Ba_{xs}$ were similar, yet concentrations in the
mesopelagic layer were smaller at Station 21, as the $Ba_{xs}$ peaked only to 524 pmol $L^{-1}$. $Ba_{xs}$ concentrations returned to
the background value at 1200 m. $Ba_{xs}$ concentration in surface waters of Station 26 were the highest of the entire
section, and reached 1888 pmol $L^{-1}$ at 50 m. Below this depth, $Ba_{xs}$ concentrations decreased back to the background
level at 100 m, then increased again, with a second peak of 451 pmol $L^{-1}$ at 200 m. In the Icelandic basin of the NADR
province, $Ba_{xs}$ concentrations were relatively high, reaching 646 and 711 pmol $L^{-1}$ at 200 m of Station 32 and at 300
m of Station 38, respectively. These stations were also characterized by $Ba_{xs}$ profiles displaying a double peak at 200
and 450 m for Station 32, and at 300 and 700 m for Station 38. Below this second maximum, $Ba_{xs}$ concentrations
decreased to the background level at 1000 m for both stations.
In the ARCT province, a similar double peak profile was observed at Station 44, in the Irminger Sea, with $Ba_{xs}$
concentrations reaching 747 pmol $L^{-1}$ at 400 m and 823 pmol $L^{-1}$ at 700 m. Then, $Ba_{xs}$ concentrations returned to the
background value at 1100 m. Close to the Greenland margin (Station 51), $Ba_{xs}$ concentrations were lower than at other
stations reaching 495 pmol $L^{-1}$ at 300 m, before to decrease until the background level at 1000 m.
The $Ba_{xs}$ concentrations of Stations 44 and 51 were compared to those obtained at Station 11 (63.5°N–324.8°E) and
Station 5 (56.9°N–317.2°E) of the GEOSECS cruise, in summer 1970 (Brewer et al., unpublished results). The $Ba_{xs}$
concentrations obtained at GEOSECS Station 11 are in similar range as those measured at GEOVIDE Station 44 (173–
658 pmol $L^{-1}$ and 116–823 pmol $L^{-1}$, respectively). Similar ranges were also observed between GEOSECS Station 5
and GEOVIDE Station 51 (170–402 pmol $L^{-1}$ and 127–359 pmol $L^{-1}$, respectively).
In the Labrador Sea (Stations 64, 69 and 77), high $Ba_{xs}$ concentrations (> 450 pmol $L^{-1}$ and up to 863 pmol $L^{-1}$ at
Station 69) were measured at greater depths than for other stations, and $Ba_{xs}$ concentrations did not return to the
background level at 1000–1500 m as observed elsewhere in the section. Samples dedicated to trace metals (Go-Flo
sampling) indicated that $Ba_{xs}$ concentrations decreased to the background level (180 pmol $L^{-1}$) at 1300, 1700 and 1200
m for Stations 64, 69 and 77, respectively.

### 3.3. Mesopelagic $Ba_{xs}$

The $Ba_{xs}$ concentrations were integrated (trapezoidal integration) over two depth intervals of the mesopelagic layer
(100–500 m and 100–1000 m; Table 2) to obtain a depth-weighted average (DWA) $Ba_{xs}$ values.
The DWA $Ba_{xs}$ values between 100 and 500 m ranged from 399 to 672 pmol $L^{-1}$ and from 315 to 727 pmol $L^{-1}$ between
100 and 1000 m (Stations 51 and 69, respectively). For both depth intervals, DWA $Ba_{xs}$ values varied by less than a
factor of 1.4, being larger for the 100–1000 m interval in the Labrador Sea (Stations 64, 69 and 77). We, thus,
considered the 100–1000 m depth interval as the interval representing the best the complete mesopelagic layer.
The lowest median DWA $Ba_{xs}$ was observed in the NADR province (403 ± 34 pmol $L^{-1}$ between 100 and 1000 m,
n=4), while the highest median DWA $Ba_{xs}$ was observed in the ARCT province (566 ± 155 pmol $L^{-1}$ between 100 and



1000 m, n=5). Station 13, in the NAST province, was characterized by a similar DWA $Ba_{xs}$ than those determined
within the NADR province, i.e. 419 pmol $L^{-1}$. In the ARCT province, the DWA $Ba_{xs}$ were variable, ranging from 315
pmol $L^{-1}$ at Station 51 to 727 pmol $L^{-1}$ at Station 69, with a high DWA $Ba_{xs}$ of 633 at Station 44.
Because of differences in the depths at which the $Ba_{xs}$ signal decreased to the background level, we decided to integrate
the $Ba_{xs}$ signal down to 1000 m to enable comparison of the DWA $Ba_{xs}$ values between stations.
In a few cases, such as the stations in the Labrador Sea where the background level was not reached at 1000 m, we
estimated the DWA $Ba_{xs}$ below 1000 m. However, this modification did not change significantly the magnitude of the
DWA $Ba_{xs}$, with the concentrations integrated over the 100–1300 m (Station 64), 100–1700 m (Station 69) and 100–
1200 m (Station 77) being between 1.0 and 1.5 fold lower than for integrations over the 100–1000 m range.
**4.    Discussion**
**4.1.    Factors influencing the DWA $Ba_{xs}$ in the North Atlantic**
**4.1.1. Influence of the intensity and stage of the bloom**
We compared our $Ba_{xs}$ inventories with the averaged biomass development from January to June 2014 (Fig. 6), which
is the period integrating the complete productive period in the North Atlantic until our sampling.
Along the GEOVIDE transect, the most productive area during this period was clearly the Labrador Sea of the ARCT
province, where Chl-*a* concentrations averaged 6 mg $m^{-3}$ (Fig. 6). This basin was sampled during the decline of the
bloom (Fig. 1; Chl-*a* concentration was > 3 mg $m^{-3}$ one month before the sampling, and low PP and nutrient
concentrations during sampling), which could explain the high DWA $Ba_{xs}$ observed in this area (Table 2). To a lesser
extent than the Labrador Sea, the West European basin of the NADR province, and in particular the area around Station
21, was characterized by an important biomass level between January and June (Fig. 6). This bloom started in May
(Fig. 1; Chl-a concentration ≈ 3 mg $m^{-3}$ one month before the sampling) and was still progressing during the sampling,
as indicated by the high PP (135 mmol $m^{-2}$ $d^{-1}$). These features can explain the lower DWA $Ba_{xs}$ observed at Station
21 (Table 2) compared to the Labrador Sea. The other stations of the NADR (Stations 26, 32 and 38) were sampled
during the bloom development (Fig. 1 and high PP reaching 174 mmol $m^{-2}$ $d^{-1}$ at Station 26 during sampling) and were
characterized by lower DWA $Ba_{xs}$ compared to other stations, suggesting a time lag between production and $Ba_{xs}$
signal. However, this was not the case for Station 44, in the Irminger Sea of the ARCT province, sampled during the
bloom (high PP, high Chl-*a* and high nutrient concentrations during the sampling period) and characterized by one of
the highest DWA $Ba_{xs}$. This high $Ba_{xs}$ inventory may reflect an important past bloom, as evidenced in fact by the
satellite Chl-*a* data (see Table 2 or Fig. 9 in Lemaitre et al., in prep.), highlighting the patchiness of the phytoplankton
blooms in this area.
As deduced by different authors, the mesopelagic $Ba_{xs}$ signal is related to a past surface production and builds up
during the growth season (Dehairs et al., 1997; Cardinal et al., 2001, 2005). The large regional and temporal variability
of the bloom development involves thus a large variability of the mesopelagic $Ba_{xs}$ signal in the North Atlantic.



### 4.1.2. Influence of water masses/physical forcing


The largest $Ba_{xs}$ inventory was determined in the Labrador Sea (Stations 64, 69 and 77), characterized by the presence
of the Labrador Sea Water (LSW; potential temperature between 2.7 and 3.8 °C and salinity below 34.9; Harvey, 1982;
Yashayaev, 2007) in the upper 1500 m. The LSW formation takes place in the central Labrador Sea, where convection
reached ~1700 m during the winter preceding GEOVIDE (Fig. 2; Kieke and Yashayaev, 2015). The deepening of the
mixed layer depth has been recently shown as a major source (from 23 % to > 100 % in high latitude regions) of
organic carbon to the mesopelagic zone (Dall'Olmo et al., 2016), supporting the carbon demand of the mesopelagic
food web (Burd et al., 2010; Aristegui et al., 2009). Moreover, the highest mesopelagic prokaryotic heterotrophic
abundance during GEOVIDE was observed in the central Labrador Sea (Station 69), reaching 896 cells $\mu L^{-1}$ at 500 m,
while the median values at the other stations for which bacterial cell numbers were available for the mesopelagic zone
(Stations 13, 21, 26, 32 and 38), reached $258 \pm 60$ cells $\mu L^{-1}$ at the same depth (J. Laroche, J. Ratten and R. Barkhouse,
personal communication). Therefore, the LSW subduction area appears to reinforce the microbial loop by increasing
the layer in which the bacteria can thrive feeding on increased food supplies. This condition appears to enhance the
$Ba_{xs}$ inventory.
The LSW was also present in the Irminger Sea between 500 and 1000 m at Station 44 (Fig. 2), where a second $Ba_{xs}$
peak was observed (823 and 632 pmol $L^{-1}$ at 700 and 800 m; Fig. 5). In the Temperature-Salinity plot, these high $Ba_{xs}$
concentrations are clearly associated with the presence of LSW (Fig. 7a), suggesting that this $Ba_{xs}$ peak could represent
an advected signal. At Station 44, the contribution of the advected signal would be about 89 pmol $L^{-1}$ (14 % of the total
signal), which is within the uncertainty of the flux. Similarly, a second $Ba_{xs}$ peak was observed at 450 m of Station 32
(Fig. 5) and the Temperature-Salinity plot (Fig. 7b) points out that this second peak was related to the presence of the
Subarctic Intermediate water (SAIW; temperature of $5.6 \pm 0.1$ °C and salinity of $34.70 \pm 0.02$; Alvarez et al., 2004),
which contributes to 14 pmol $L^{-1}$ (3 % of the total signal).
Station 38 was also characterized by a second $Ba_{xs}$ peak at 700 m (Fig. 5), probably unrelated to the presence of a
specific water mass since there are no changes in the Temperature-Salinity plot (Fig. 7b). No water masses influence
was detected at the remaining stations. Overall, lateral transport influencing the local $Ba_{xs}$ distributions was observed
at two stations during GEOVIDE but these did not significantly modify the magnitude of the local mesopelagic $Ba_{xs}$
inventory. However, the subduction occurring in the Labrador Sea resulted in larger mesopelagic DWA $Ba_{xs}$, probably
due to high organic export and associated prokaryotic heterotrophic activity in these areas.

### 4.1.3. Influence of the phytoplankton community structure


The different $Ba_{xs}$ inventories may also be influenced by the different phytoplankton communities encountered in each
province.
The ARCT province was dominated by diatoms (median value: $63 \pm 19$ % of the total phytoplankton community taxa;
Tonnard et al., 2017, this issue) and was characterized by the highest DWA $Ba_{xs}$ values while the NAST and NADR
provinces were characterized by higher abundance of haptophytes (median value: $43 \pm 16$ % of the total phytoplankton
community taxa; Tonnard et al., 2017, this issue) and by lower $Ba_{xs}$ inventories. Coccolithophorids are part of the
haptophyte family and their dominance was confirmed by visual observations on filters (surface down to 400 m) by
FE-SEM. Calcifiers, such as coccolithophorids, have been shown to be more efficient in transferring carbon to the



deep ocean compared to diatoms (Francois et al., 2002; Klaas and Archer, 2002; Lam et al., 2011). This difference
could result from the low compaction or the high fluffiness of diatom aggregates, the high degree of degradability of
organic compounds within diatom aggregates, the greater calcite density, the resistance of calcite to grazing and the
more refractory nature of the exported organic matter associated to calcite (Bach et al., 2016; Francois et al., 2002;
Klaas and Archer, 2002; Lam et al., 2011; Le Moigne et al., 2013a; Ragueneau et al., 2006). Therefore, enhanced
particle degradation when diatoms are predominant seems to increase the mesopelagic DWA $Ba_{xs}$.

### 4.2. Relationship between $Ba_{xs}$ and carbon remineralisation in the North Atlantic

The mesopelagic $Ba_{xs}$ inventory can be related to the rate of oxygen consumption ($JO_2$), which can be then converted
into a mesopelagic carbon remineralisation (Eq. 2). The relationship between the $Ba_{xs}$ inventory and the $JO_2$ has been
determined in the Southern Ocean (Shopova et al., 1995; Dehairs et al., 1997; Eq. 1), and it is of interest to investigate
if this relationship can be applied in the North Atlantic.
For this purpose, we have calculated the oxygen utilization rate (OUR; $\mu mol\ kg^{-1}\ yr^{-1}$), which is determined by dividing
the apparent oxygen utilization (AOU, in $\mu mol\ kg^{-1}$) by the water mass age (Table S1). From the Iberian coasts to
Greenland, the age calculation was based on CFC-12 (when available, otherwise CFC-11) determined in 2012 (OVIDE
CARINA cruise, de la Paz et al., 2017). In the Labrador Sea, the mean age of LSW has been estimated by Rhein et al.
(2015) based on a 25 year record of CFC contents. The OUR was then integrated over the 100–1000 m layer. The
relationships between the $Ba_{xs}$ inventories and the oxygen consumption rates determined using the Southern Ocean
equation (Eq. 1) and the one deduced here for the North Atlantic (via the OUR) are compared in Figure 8. This new
relationship is significant (p-value = 0.006; Fig. 8) but does not include the Station 44, which was located in the
Irminger Gyre (Zunino et al., 2017; this issue). This physical feature may reflect a greater and/or longer mesopelagic
$Ba_{xs}$ accumulation in this area explaining the difference compared to other stations.
Figure 8 suggests that for a given mesopelagic $Ba_{xs}$ inventory the $JO_2$ is smaller in the North Atlantic than in the
Southern Ocean, with the relationship for the North Atlantic being:
$$JO_2 = (\text{mesopelagic } Ba_{xs} - Ba_{\text{residual}}) / 24000 \qquad (4)$$
where $JO_2$ is the rate of oxygen consumption ($\mu mol\ L^{-1}\ d^{-1}$), *mesopelagic $Ba_{xs}$* is the depth-weighted average in the
mesopelagic layer (DWA; $pmol\ L^{-1}$), $Ba_{residual}$ is the deep ocean $Ba_{xs}$ value observed at zero oxygen consumption (or
$Ba_{xs}$ background signal; 250 $pmol\ L^{-1}$).
However, this new relationship is sensitive to potential errors. The OUR has been shown to under-estimate the ocean
respiration because of the non-proportional diffusive mixing of AOU and water mass age resulting in an excess loss
of AOU versus age (Koeve and Kähler, 2016). This would decrease the mismatch between the Southern Ocean and
North Atlantic regressions. Errors can also be directly associated to the CFC-based age values of the water masses,
which would appear especially critical for LSW. Indeed, the severe winter preceding the cruise (2013/2014) appeared
to have strongly ventilated LSW with a mixed layer depth exceeding 1700 m (Kieke and Yashayaev, 2015), indicating
that the mean age (4 years) estimated by Rhein et al. (2015) may have over-estimated the real LSW age (P. Lherminier,
personal communication). Moreover, in the Labrador Sea, the residence time of LSW strongly varies between the
central Labrador Sea (4–5 years) and the boundary currents off the Greenland and Newfoundland coasts (a few months;



Deshayes et al., 2007; Straneo et al., 2003). The over-estimation of these ages could have directly under-estimate the
OUR, resulting, again, to bring more closely together the both North Atlantic and Southern Ocean regressions.
In the following discussion, carbon remineralisation fluxes deduced in the North Atlantic (GEOVIDE and GEOSECS
cruises) are estimated by using Eq. (4) and (2). These fluxes are compared to carbon mesopelagic remineralisation
fluxes calculated with the $Ba_{xs}$ method in the World Ocean (Table 3). Fluxes obtained during GEOVIDE, especially
in the ARCT province, were higher than fluxes based on $Ba_{xs}$ data reported for the Southern and Pacific Oceans,
highlighting an important remineralisation in the northern part of this basin compared to other oceans.

### 4.3. Comparison of remineralisation fluxes deduced from different methods

#### 4.3.1. Remineralisation from direct measurements

In the North Atlantic, carbon respiration rates were also deduced by surface drifting sediment traps and associated-
shipboard incubations. Collins et al. (2015) determined very high respiration rates reaching 39 and 72 mmol C m$^{-2}$ d$^{-1}$
at sites located in the NADR and in the ARCT provinces, respectively. Nevertheless, these high fluxes were deduced
in the upper mesopelagic layer (50–150 m) where respiration is greater compared to the lower mesopelagic layer (150–
1000 m). This different depth interval could thus explain the lower integrated respiration rates determined in our study.
Using a similar method supplemented by measurements of zooplankton respiration, Giering et al. (2014) determined
respiration rates in the NADR province (PAP site) reaching 7.1 mmol C m$^{-2}$ d$^{-1}$ during summer. This flux, determined
over the 50–1000 m depth interval, is in the same order of magnitude than our estimates in the NADR province.

#### 4.3.2. Remineralisation from the deep sediment traps

The remineralisation flux in the mesopelagic layer can also be derived from the difference between a deep POC export
flux and a surface POC export flux. Honjo et al. (2008) compiled deep POC export fluxes from moored and time-series
sediment traps and calculated the corresponding export production (upper-ocean POC export flux) using an ecosystem
model (Laws et al., 2000) for most world provinces. Then, by difference, the authors estimated an annual average of
carbon remineralisation fluxes in the mesopelagic layer reaching, after conversion into daily average fluxes, values of
34 mmol C m$^{-2}$ d$^{-1}$ in the ARCT province, 9 mmol C m$^{-2}$ d$^{-1}$ in the NADR province and 4 mmol C m$^{-2}$ d$^{-1}$ in the NAST
province. Noteworthy, the flux in the ARCT province was the highest mesopelagic remineralisation flux estimated
worldwide (with the region around Cape Verde), confirming the important remineralisation in the northern part of the
North Atlantic compared to other oceans. The values published by Honjo et al. (2008) for the North Atlantic are
relatively similar to the median values obtained during GEOVIDE in each province. Indeed, mesopelagic
remineralisation fluxes based on the $Ba_{xs}$ proxy at Station 13 were similar than the value reported by Honjo et al.
(2008) for the NAST province, while they were 2 and 4 fold lower in the NADR and in the ARCT provinces,
respectively.
Overall, the remineralisation fluxes deduced from the $Ba_{xs}$ proxy are in concordance with those obtained by the other
methods, confirming the order of magnitude of the mesopelagic remineralisation fluxes determined in this study of the
North Atlantic (Fig. 9).





### 4.4. The biological carbon pump in the North Atlantic

In order to investigate the efficiency of the biological carbon pump in the North Atlantic, we examined the daily PP (A. Roukaerts, D. Fonseca Batista and F. Deman, unpublished data), the upper-ocean POC export (Lemaitre et al., in prep) and the POC remineralisation in the mesopelagic layer (Table 3; Fig. 10).

During GEOVIDE, low ($\leq 12$ %) export efficiencies (i.e., the ratio between PP and POC export) were observed at most stations indicating an accumulation of biomass in surface waters or a strong turn-over of the exported organic matter due to important remineralisation occurring in the upper water column (< 100 m). Yet, relatively high POC remineralisation fluxes were also measured in the mesopelagic layer, equalling or exceeding the POC export fluxes at some stations. This highlights a strong mesopelagic remineralisation with little or no material left for export to the deep ocean, but above all, it involves an imbalance between carbon supplies and mesopelagic remineralisation.

This imbalance can be caused by the distinct time scales over which the PP, POC export and POC remineralisation fluxes operate. Indeed, the measurements of PP represent a snapshot (24h incubations) while measurements of export ($^{234}$Th) integrate several weeks and remineralisation ($Ba_{xs}$) integrate probably much longer time scales. Moreover, previous studies in the Southern Ocean showed that mesopelagic processing of exported organic carbon, as reflected by $Ba_{xs}$, has a phase lag relative to the upper-ocean processes (Dehairs et al., 1997; Cardinal et al., 2005). Thus, we do not expect mesopelagic $Ba_{xs}$ to be in phase with coinciding amplitude of PP and subsequent export. Because of the observed high remineralisation fluxes relative to the export fluxes, particularly in the ARCT province, we suppose that the surface particulate organic matter sank and accumulated in the mesopelagic layer in a period preceding the specific time windows for POC export and PP, leading to an important remineralisation. This can be amplified by the spatial and temporal variability of the phytoplankton bloom in this province, generating sudden high export events and associated remineralisation. Conversely, a fraction of POC, reaching 50% at Station 32, escape remineralisation in the NADR province. The more efficient POC transfer through the mesopelagic layer of this province may be explained by the early sampling compared to the bloom development and/or by the presence of calcified phytoplankton species (see Section 4.1).

Overall, the remineralisation in the mesopelagic layer is an important process that needs to be taken into account as our results point to the poor ability of specific areas within the North Atlantic to sequester carbon at depth below 1000 m in spring 2014.

## 5. Conclusion

We investigated mesopelagic carbon remineralisation fluxes in the North Atlantic during the spring 2014 (GEOVIDE section) using for the first time the particulate biogenic barium inventories measured for this area. The excess barium ($Ba_{xs}$) content in the mesopelagic layer varied between the different provinces of the North Atlantic. The highest $Ba_{xs}$ inventory was observed in the ARCT province, where high carbon production rates were observed earlier in the season. The regional variations of the $Ba_{xs}$ inventory can also be due to the different phytoplankton community composition encountered along this trans-Atlantic section. Lower contents were determined where the smaller calcified phytoplankton species dominated, as in the NADR province. Finally, the ARCT province was also characterized by an important water mass subduction, generating a larger transport of organic matter to the deep ocean, which might have resulted into an important $Ba_{xs}$ accumulation in the mesopelagic layer.





Using the OUR method, we confirmed that mesopelagic $Ba_{xs}$ content can be related to an oxygen consumption, but the
relationship between both parameters slightly changed in comparison to the relationship proposed elsewhere for the
Southern Ocean. A new relationship is thus proposed for the North Atlantic. This proxy approach provided similar
estimations of remineralisation fluxes obtained by independent methods (moored sediment traps, incubations) in the
North Atlantic.
Overall, in spring 2014, the remineralisation was equal or larger than POC export in the subtropical and subpolar
provinces of the North Atlantic, highlighting the important impact of the mesopelagic remineralisation on the
biological carbon pump and indicating that little to no POC was transferred below 1000 m in this region.
**Acknowledgements**
We would like to thank the captain and the crew of the R/V Pourquoi Pas?, as well as Fabien Perault and Emmanuel
De Saint Léger from the CNRS DT-INSU for their help during the CTD deployments. Pierre Branellec, Floriane
Desprez de Gésincourt, Michel Hamon, Catherine Kermabon, Philippe Le Bot, Stéphane Leizour and Olivier Ménage
are also acknowledged for their technical help during the cruise. We acknowledge Lorna Foliot, Raphaëlle Sauzède,
Joséphine Ras, Hervé Claustre and Céline Dimier for the sampling and analysis of pigments. We would like to thank
the co-chief scientists Pascale Lherminier and Géraldine Sarthou. We also express our thanks to Pascale Lherminier,
Herlé Mercier, Monika Rhein, Julie Deshayes and Claude Talandier for providing useful help on the characteristics of
the Labrador Sea Water. Satellite chlorophyll-a data and visualizations used in this study were produced with the
Giovanni and the Ocean Color (Ocean Biology Processing Group; OBPG) online data system, developed and
maintained by the NASA.
This work was funded by the Flanders Research Foundation (project G071512N), the Vrije Universiteit Brussel
(strategy research program: project SRP-2), the French National Research Agency (ANR-13-BS06-0014 and ANR-
12-PDOC-0025-01), the French National Center for Scientific Research (CNRS-LEFE-CYBER), IFREMER and the
"Laboratoire d'Excellence" Labex-Mer (ANR-10-LABX-19).

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





**Table 1:** **Particulate Barium (Ba) and Aluminium (Al) concentrations and resulting recoveries (bold and italic percentages)**
**of the certified reference materials SLRS-5 (river water), BHVO-1 (basalt powder), JB-3 (basalt powder) and JGb-1 (gabbro**
**powder).**

|  | Ba | Al |
|---|---|---|
| **SLRS-5** (µg kg$^{-1}$) | 13 ± 1 | 47 ± 2 |
| n=4 | 95 % | 95 % |
| **BHVO-1** (µg g$^{-1}$) | 129 ± 1 | 70118 ± 984 |
| n=4 | 93 % | 96 % |
| **JB-3** (µg g$^{-1}$) | 229 ± 13 | 92144 ± 1620 |
| n=4 | 94 % | 101 % |
| **JGb-1** (µg g$^{-1}$) | 68 ± 15 | 91491 ± 732 |
| n=4 | 106 % | 99 % |






















**Table 2: Depth-weighted average (DWA) values of mesopelagic Ba$_{xs}$ (in pmol L$^{-1}$) integrated between 100–500 m and 100–**
**1000 m depths. The biogeochemical provinces defined by Longhurst et al. (1995) are also indicated: NAST: North Atlantic**
**subtropical gyre; NADR: North Atlantic drift; ARCT: Atlantic Arctic.**


| Province | Station | Latitude (° N) | Longitude (° E) | DWA Ba$_{xs}$ 100-500 m (pmol L$^{-1}$) | | | DWA Ba$_{xs}$ 100-1000 m (pmol L$^{-1}$) | | |
|---|---|---|---|---|---|---|---|---|---|
| NAST | 13 | 41.4 | -13.9 | 578 | ± | 89 | 419 | ± | 71 |
| | 21 | 46.5 | -19.7 | 428 | ± | 69 | 394 | ± | 64 |
| NADR | 26 | 50.3 | -22.6 | 405 | ± | 59 | 391 | ± | 58 |
| | 32 | 55.5 | -26.7 | 522 | ± | 81 | 413 | ± | 66 |
| | 38 | 58.8 | -31.3 | 572 | ± | 86 | 465 | ± | 78 |
| | 44 | 59.6 | -38.9 | 678 | ± | 104 | 633 | ± | 98 |
| | 51 | 59.8 | -42 | 399 | ± | 72 | 315 | ± | 58 |
| ARCT | 64 | 59.1 | -46.1 | 464 | ± | 95 | 566 | ± | 99 |
| | 69 | 55.8 | -48.1 | 672 | ± | 111 | 727 | ± | 118 |
| | 77 | 53 | -51.1 | 472 | ± | 80 | 505 | ± | 83 |







**Table 3: Comparison of the Ba$_{xs}$ inventory (pmol L$^{-1}$) and related-carbon mesopelagic remineralisation fluxes (mmol C m$^{-2}$ d$^{-1}$) obtained in the world's ocean.**

| Cruise (season) | Location | Features | depth interval, m | DWA Ba$_{xs}$ pmol L$^{-1}$ | MR fluxes mmol C m$^{-2}$ d$^{-1}$ | Reference |
|---|---|---|---|---|---|---|
| CLIVAR SR3 - SAZ298 (spring/summer) | Southern Ocean | spring<br>summer | 150 - 400 | 235 - 554<br>296 - 353 | 0.3 - 3.0<br>0.2 - 3.4 | Cardinal et al., 2005 |
| VERTIGO (summer) | Pacific Ocean | oligotrophic (Aloha station)<br>mesotrophic (K2 station) | 150 - 500 | 157 - 205<br>367 - 713 | 1.0 - 3.0<br>2.7 - 8.8 | Dehairs et al., 2008 |
| EIFEX (summer) | Southern Ocean | fertilized (in patch)<br>HNLC (out patch) | 150 - 1000 | 273 - 415<br>233 - 423 | 2.6 - 7.7<br>1.2 - 8.0 | Jacquet et al., 2008a |
| KEOPS (summer) | Southern Ocean | fertilized (A3 station)<br>HNLC (C11 station) | 125 - 450 | 342 - 401<br>309 - 493 | 2.1 - 2.8<br>1.7 - 4.0 | Jacquet et al., 2008b |
| SAZ-SENSE (summer) | Southern Ocean | fertilized (SAZ east)<br>HNLC (SAZ west) | 100 - 600 | 244 - 395<br>199 - 249 | 3.0 - 6.1<br>2.1 - 3.1 | Jacquet et al., 2011 |
| Bonus GoodHope (summer) | Southern Ocean | North of PF<br>South of PF | 125 - 600 | 284 - 497<br>235 - 277 | 2.1 - 6.4<br>1.1 - 1.9 | Planchon et al., 2013 |
| KEOPS 2 (spring) | Southern Ocean | fertilized (A3 station)<br>HNLC (R2 station) | 150 - 400 | 267 - 314<br>572 | 0.9 - 1.2<br>4.2 | Jacquet et al., 2015 |
| GEOSECS II (summer) | North Atlantic | NAST+NADR<br>ARCT | 100 - 1000 | 199 - 361<br>242 - 413 | 0.5 - 4.9<br>1.7 - 6.3 | Brewer (unpublished values) |
| **GEOVIDE (spring)** | **North Atlantic** | **NAST (station 13)** | **100 - 1000** | **419** | **4.6** | **this study** |
| | | **NADR (station 21)** | | **394** | **3.9** | |
| | | **NADR (station 26)** | | **391** | **3.8** | |
| | | **NADR (station 32)** | | **413** | **4.4** | |
| | | **NADR (station 38)** | | **465** | **5.9** | |
| | | **ARCT (station 44)** | | **633** | **10** | |
| | | **ARCT (station 51)** | | **315** | **1.8** | |
| | | **ARCT (station 64)** | | **566** | **8.6** | |
| | | **ARCT (station 69)** | | **727** | **13** | |
| | | **ARCT (station 77)** | | **505** | **6.9** | |





**Table 4:** **Comparison of the mesopelagic POC remineralisation fluxes (Remineralisation) with primary production (PP)**
**and POC export fluxes in the upper water column (Export). All fluxes are expressed in mmol C m$^{-2}$ d$^{-1}$. [1] PP data from**
**A. Roukaerts, D. Fonseca Batista and F. Deman (unpublished data); [2] Export data from Lemaitre et al. (in prep.).**

| Station | ARCT - Labrador Sea | | | ARCT - Irminger Sea | | | NADR | | | NAST |
|---|---|---|---|---|---|---|---|---|---|---|
| | 77 | 69 | 64 | 51 | 44 | 38 | 32 | 26 | 21 | 13 |
| PP [1] | 95 | 31 | 67 | 165 | 137 | 68 | 142 | 174 | 135 | 80 |
| Export [2] | 6 | 10 | 8 | 3 | 1 | 5 | 8 | 7 | 5 | 2 |
| Remineralisation | 7 | 13 | 9 | 2 | 10 | 6 | 4 | 4 | 4 | 5 |









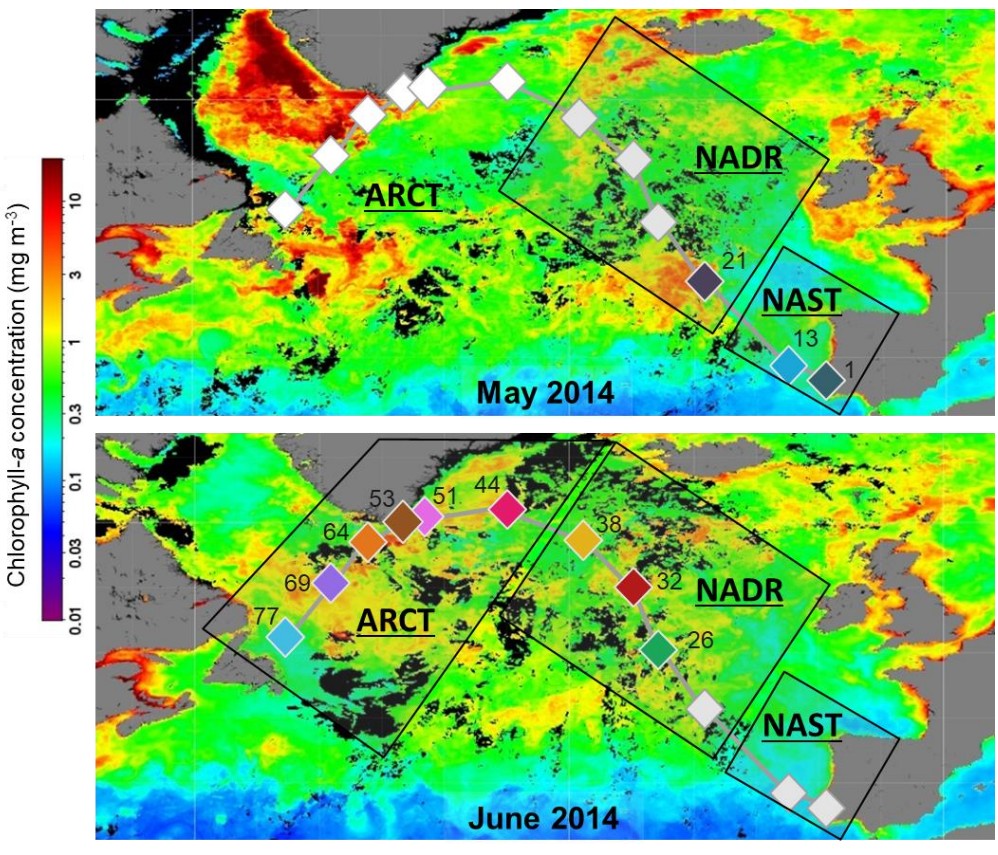


**Figure 1:** Satellite Chlorophyll-a concentrations (MODIS Aqua from http://oceancolor.gsfc.nasa.gov), in mg m$^{-3}$ during the GEOVIDE cruise (May and June 2014). The province are indicated: NAST: North Atlantic Subtropical gyre; NADR: North Atlantic Drift; ARCT: Atlantic Arctic. Diamonds indicate stations, coloured according to their approximate time of sampling.














**(a)**

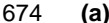

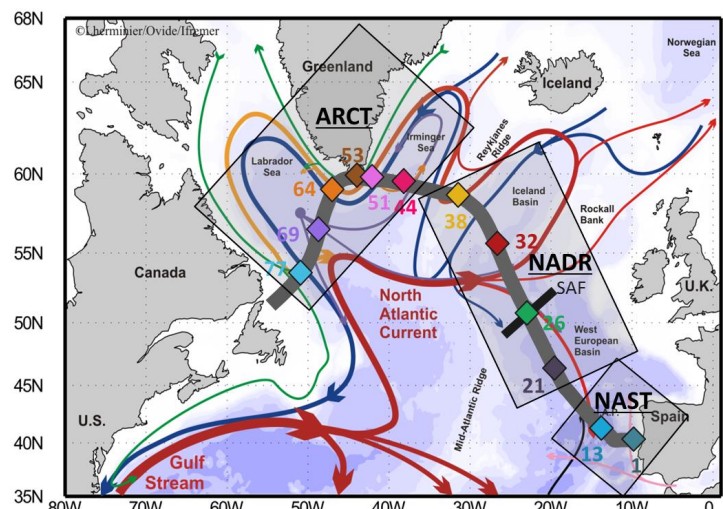


**(b)**

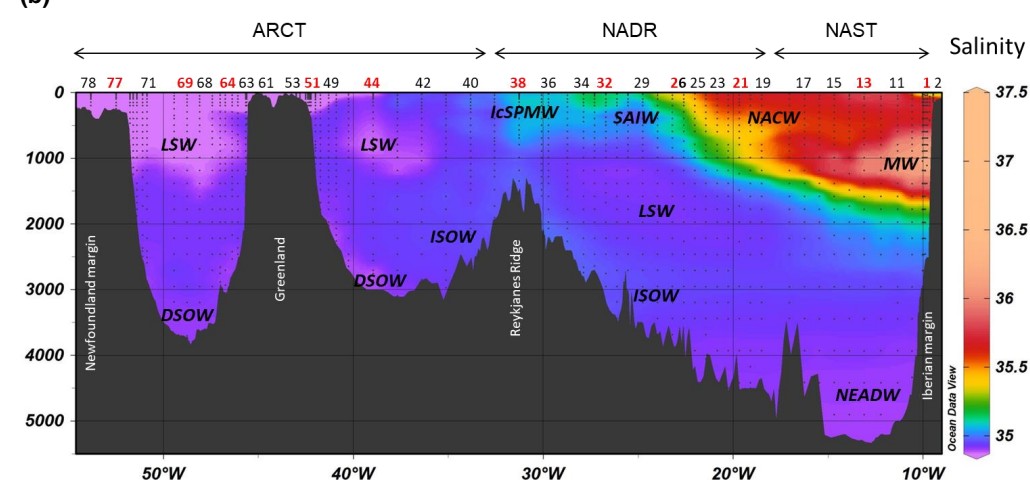


**Figure 2: (a) Schematic of the circulation features, adapted from García-Ibáñez et al. (2015). Bathymetry is plotted in**
**colour with colour change at 100 m, at 1000 m and every 1000 m below 1000 m. The red and green arrows represent**
**the main surface currents, the pink and orange arrows represent the intermediate currents and the blue and purple**
**arrows represent the deep currents. (b) Salinity along the GEOVIDE section, and associated water masses: LSW:**
**Labrador Sea Water; ISOW: Iceland–Scotland Overflow Water; IcSPMW: Iceland Subpolar Mode Water; SAIW:**
**Subarctic Intermediate Water; NACW: North Atlantic Central Waters; MW: Mediterranean Water; DSOW: Denmark**
**Strait Overflow Water; NEADW: North East Atlantic Deep Water. Stations in red correspond to stations where detailed**
**vertical Niskin profiles were collected.  In both figures, the provinces are also indicated: NAST: North Atlantic**
**Subtropical gyre; NADR: North Atlantic Drift; ARCT: Atlantic Arctic.**





**Figure 3: Barite particles observed by FE-SEM at (a) Station 38 (300 m); (b) Station 44 (700 m); (c and d) Station 69 (600 m). White arrows indicate the position of barite crystals.**




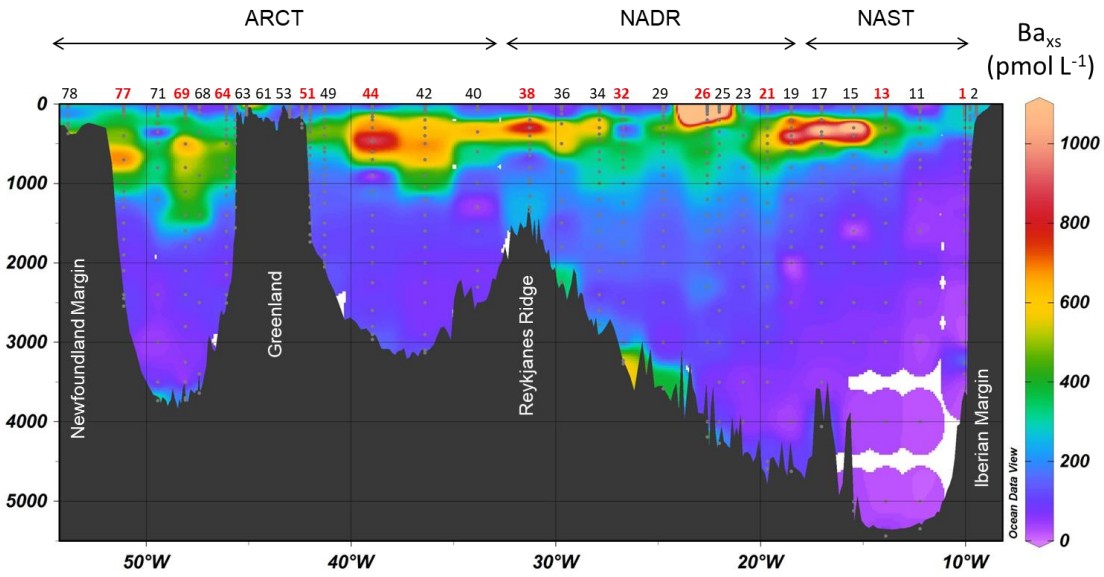

**Figure 4:** **Section of the particulate biogenic barium (Ba$_{xs}$) in pmol L$^{-1}$ determined in samples collected with the Go-Flo**

**bottles. Stations in red are those where profiles were obtained from Niskin-rosette sampling.**







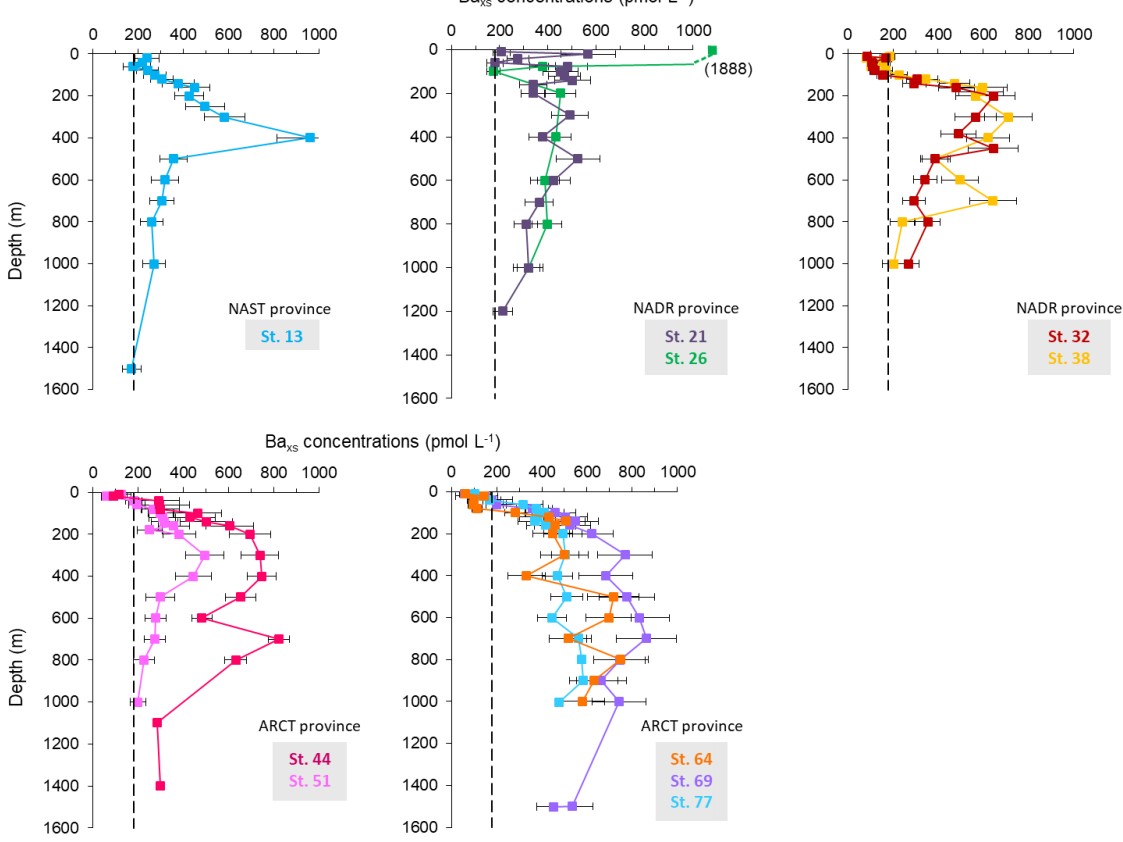

**Figure 5:** Individual profiles of Ba$_{xs}$ concentrations (in pmol L$^{-1}$) determined using Niskin bottles.





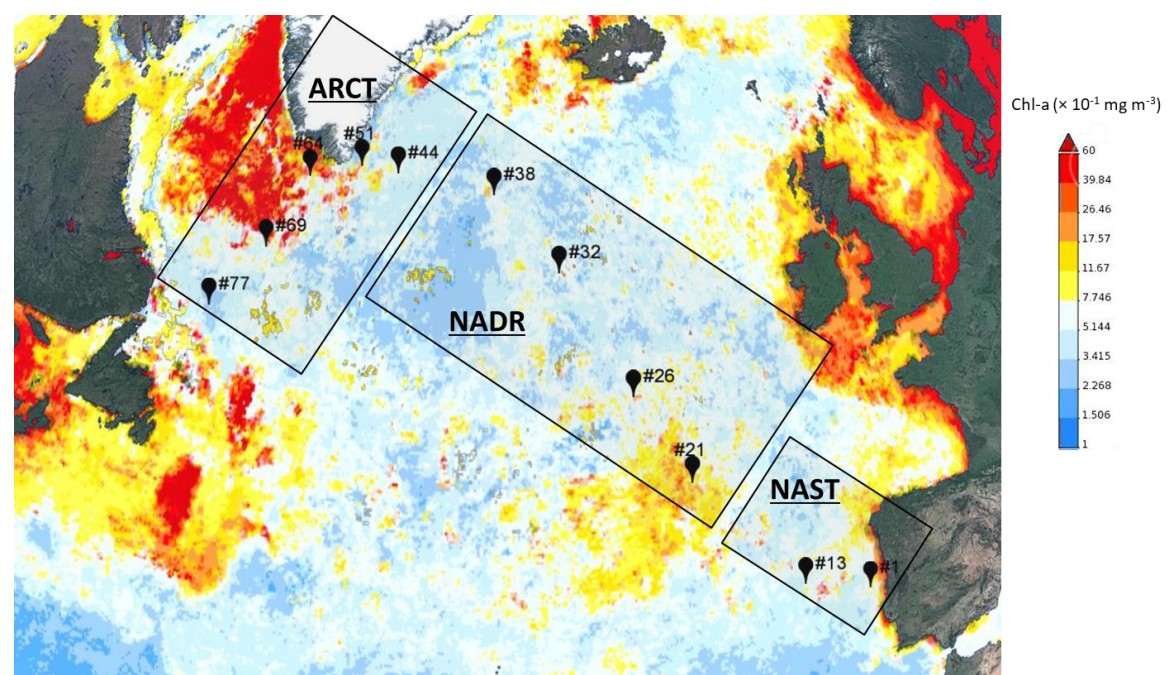


**Figure 6**: Time averaged map of Chlorophyll-a concentrations (in mg m⁻³) over January–June 2014 (monthly 4 km MODIS
Aqua model; http://giovanni.sci.gsfc.nasa.gov/).











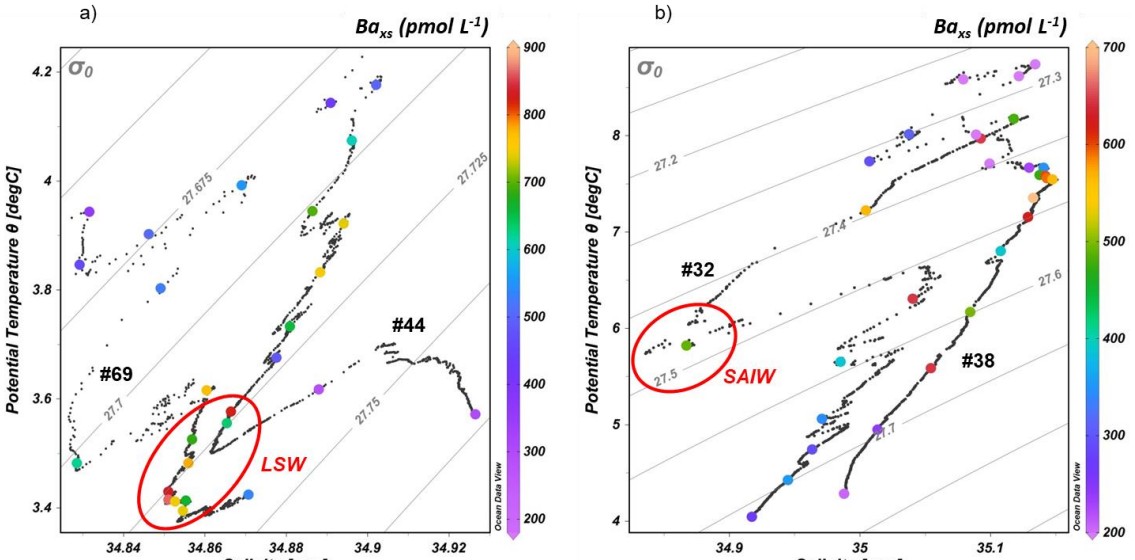


**Figure 7:** Potential temperature θ - salinity plots for the Stations (a) #44 and #69 and (b) #32 and #38 of the GEOVIDE cruise focus on the 50–2000 m depth interval. The concentrations of Ba$_{xs}$ are shown by the coloured points. Isopycnals are also represented. LSW: Labrador Sea Water; SAIW: Subarctic Intermediate Water.










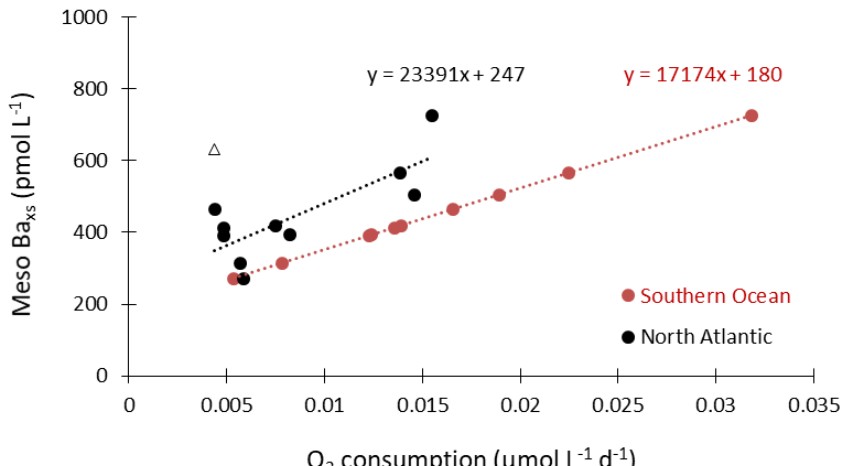


**Figure 8:** Regression of mesopelagic Ba$_{xs}$ (pmol L$^{-1}$) versus O$_2$ consumption rate (μmol L$^{-1}$ d$^{-1}$) using the Southern Ocean transfer function from Dehairs et al. (1997; red dots) and the transfer function obtained here for the North Atlantic (black dots). Station 44 (triangle) was excluded from the regression.








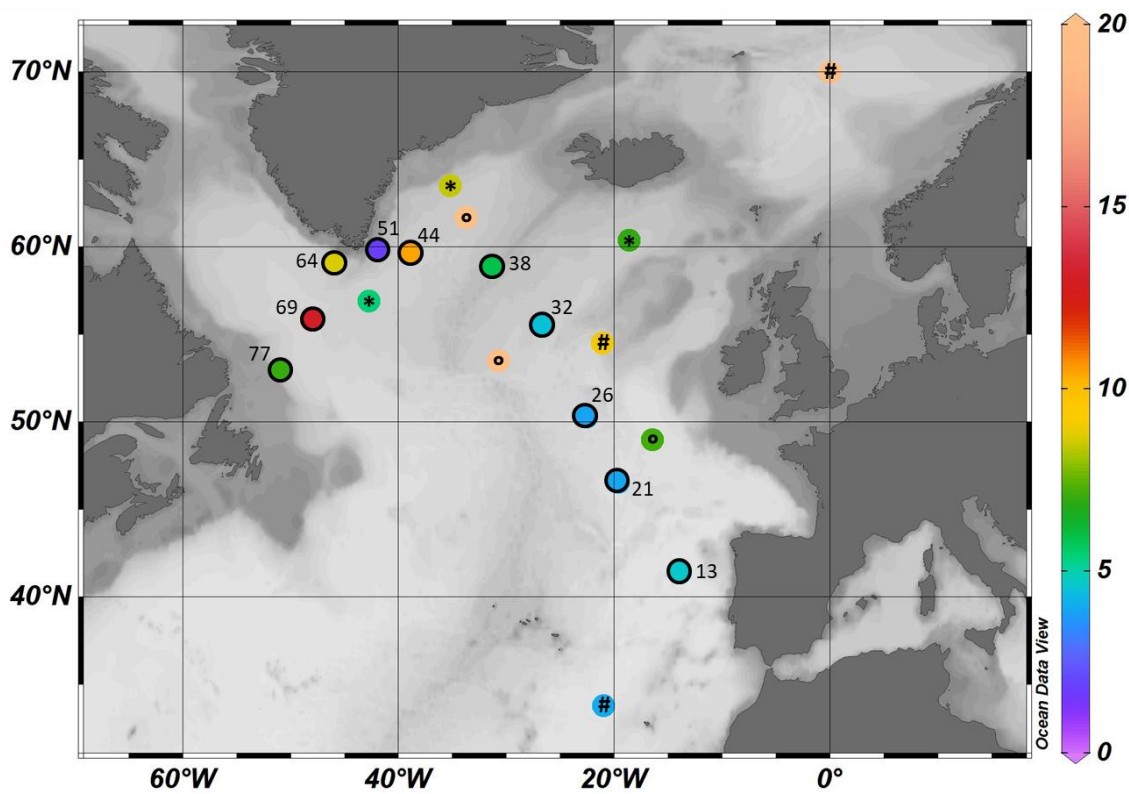


**Figure 9:** **Comparison of POC remineralisation fluxes from this study (circles lined in black) to remineralisation fluxes from literature in the North Atlantic. Note that the cited literature studies used different methods for determining remineralisation fluxes: moored sediment traps (# symbols, Honjo et al., 2008), on-board incubations (° symbols, Giering et al., 2014; Collins et al., 2015); excess barium proxy (* symbols, Brewer et al., unpublished results).**




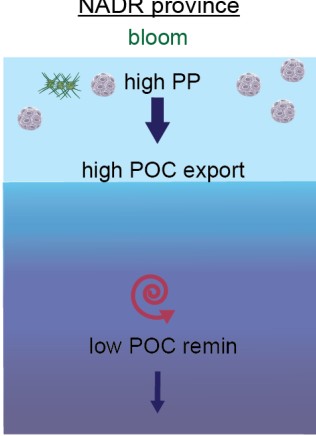
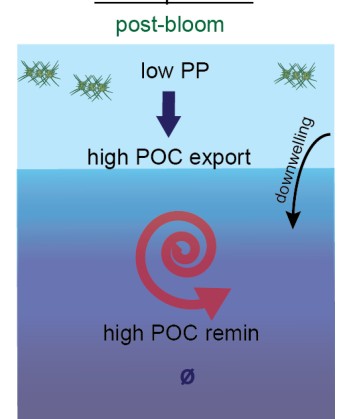
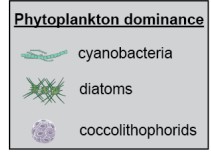

739

740

**Figure 10:** **General schematic of the biological carbon pump during GEOVIDE in the NAST, NADR and ARCT provinces.**
**Primary production (PP) data from A. Roukaerts and D. Fonseca Batista (unpublished data); particulate organic carbon**
**(POC) export fluxes from Lemaitre et al. (in prep.); and POC remineralisation fluxes from this study. The dominating**
**phytoplankton communities and the stage of the bloom are also indicated.**

745