# Peer review of "Particulate barium tracing significant mesopelagic carbon remineralisation in the North Atlantic"

_Biogeosciences, 2017_

## Referee Comment (RC1) · Anonymous Referee #1 · 16 Dec 2017

Review of the manuscript titled *Particulate barium tracing significant mesopelagic carbon remineralisation in the North Atlantic* by Lemaitre et al. 2017.

Reviewer: Anonymous

**General comments**

The study traces mesopelagic (100-1000 m) remineralisation of particulate organic carbon (POC) in four different provinces of the North Atlantic using the excess of Barium ($Ba_{ex}$) as a proxy. This technique has been successfully used to quantify remineralisation in the Southern Ocean, but for the first time applied in the North Atlantic.

The study reveals important regional variations in the magnitude of mesopelagic carbon remineralisation and thus the ability of the North Atlantic to sequester atmospheric $CO_2$. Based on the measured $Ba_{ex}$ concentrations, the mesopelagic POC remineralisation is the highest in the sub-polar North Atlantic, while in the temperate North Atlantic more POC sinks below 1000 m depth. This work also shows that in some regions mesopelagic POC remineralisation flux exceeds downward POC flux from the upper ocean, thereby highlighting the significance of the remineralisation for the strength and efficiency of the biological carbon pump.

The authors provide strong supporting arguments and evidence from multiple sources to interpreted the major drivers of the observed regional differences in mesopelagic $Ba_{ex}$ inventory and hence POC remineralisation.

-   The authors use satellite observations of surface Chl-a concentration over relevant period and areas to demonstrate that intensity and stage of phytoplankton bloom relates to the variability of mesopelagic $Ba_{ex}$ inventory and hence, the magnitude of POC remineralisation.
-   By tracking the distribution of the water masses in the sampling region, the authors find that in the subpolar region, the deepening of the mixed layer not only leads to a higher POC supply to depth but also reinforces microbial loop as bacteria have more POC to thrive on. As a result, mesopelagic $Ba_{ex}$ inventory is very high in these regions. The authors also show that at some stations $Ba_{ex}$ peak at depth can be of allochthonous and advected from the regions of high POC remineralisation.
-   Finally, the mesopelagic $Ba_{ex}$ inventory was shown to be related to the phytoplankton community structure. Regions dominated by calcifying phytoplankton had smaller $Ba_{ex}$ in the mesopelagic zone (= smaller remineralisation), while diatom-dominated provinces had very large $Ba_{ex}$ between 100-1000 m.

The authors also show that their Ba-derived POC remineralisation fluxes are comparable to the previous records from direct measurements of carbon respiration rates and estimates from sediment trap fluxes. This confirms high quality of the data presented in this study.

The development of the relationship between $Ba_{ex}$ concentrations and oxygen consumption specifically for the North Atlantic expands the application of Ba method for tracing remineralisation and adds significantly to originality and importance of the Lemaitre at al.'s work.

Overall, the manuscript represents a very neat, well-structured, mature piece of work. The sampling and analytical methods implemented are robust, while results, discussion and conclusions are very clearly presented and easy to follow. I recommend the publication of this manuscript in the current

form, although the authors may take up some of my minor suggestion for the final version of the manuscript. Those are listed below:

Lines 87-92: This sentence is too long and therefore confusing. I suggest splitting it into two shorter sentences.

Line 93 and 95: Please state the depth of the upper ocean POC export flux

Line 96: For the transfer efficiency, I suggest to replace 'e.g.' with 'here defined as'; again, please state the depth of upper ocean POC export flux

Lines 228-232: The comparison of the $Ba_{ex}$ inventory at stations 44 and 51 to the GEOSECS $Ba_{ex}$ concentration data would benefit from adding the latter to the respective individual profiles in Figure 5.

Lines 334-336: What are the $R^2$ and $p$ values for the new relationship between Ba and oxygen consumption with and without station 44? Both $R^2$ and $p$ values should feature the regression in Figure 8.

Lines 649: Table 3 shows a comparison of $Ba_{ex}$ inventory and related POC remineralisation fluxes in the global ocean. The manuscript will benefit from having MR fluxes plotted on a global map.

Line 705: The legend of Figure 5 should include the reference to the dashed line.

Lines 735-738: The legend of Figure 9 should acknowledge the use of Ocean Data View (Schlitzer et al. 2004).

---

## Referee Comment (RC2) · Anonymous Referee #2 · 26 Dec 2017

In this manuscript, Lemaitre and coauthors presented Baxs-derived mesopelagic oxygen consumption and carbon remineralization fluxes along the GEOVIDE section in the North Atlantic. Mesopelagic carbon remineralization fluxes were further compared with those from drifting sediment traps (both shallow and deep) and shipboard incubation from previous studies. A synthesis discussion on the difference in the primary production, upper-ocean POC export and mesopelagic remineralization was probably my favorite part of this manuscript. Overall, the data presented are very interesting and contribute to further our knowledge on the biological carbon pump efficiency. However, the current manuscript can be significantly improved to avoid the redundancy throughout the text, to avoid heavy citation of data that are currently not available, and to make

clear justification of some of their conclusions.

Major comments: (1) Introduction This section is unsatisfyingly short in my opinion. It really only includes 12 lines of background (L41 – 53). It needs to be expanded to include our current knowledge of the biological carbon pump (BCP) to date: why is it important to study BCP? Why North Atlantic? Why use the Baxs proxy? What is the driving hypothesis and significance of your research?

(2) Sampling and analyses It is unclear why two completely different sampling and digesting methods were carried out for Baxs. While the authors gave great details on how different the methods were, there is little discussion on the comparison except in L207-210. The authors suggested good agreement between both datasets (their Fig S1); however, a closer look at the data only show good agreement for Baxs < 400 pmol/L. At Baxs > 400 pmol/L, data in Fig S1 (Go-Flo Baxs vs. Niskin Baxs) are more scattered. These are actually the samples that are from depths of interests (100 -1000 m), and likely suggest a discrepancy between the Go-Flo samples and Niskin ones. My other curiosity is, despite two different sampling techniques, why the authors applied two completely different filter digestions. Wouldn't it be better to use the same chemical protocols for a better data comparison?

(3) Section 3.1 This section involves large amount of discussion and I suggest moving most of this section to Section 4. I have a couple thoughts about this section. Firstly, Fig. 3 does not show any biogenic fragments, and thus I cannot judge whether barite crystals were actually observed adjacent to biogenic fragments. Secondly, barite crystals are believed to form in the microenvitonment of decaying organic matter, so it is probably expected that no barite crystal is seen in surface samples.

(4) Section 3.3 This section is difficult to read, as there is a lot of jumping back and forth between provinces. I would suggest the authors to describe their data in a consistent way. For the selection of background level depth, the authors need to justified whether the absolute background values are more important for data comparison, or whether it

is better to compare data consistently at the same depth.

(5) Section 4.2 The relationship of Baxs and carbon remineralization and their derived correlation in the North Atlantic are shown here, and the comparison between these and those in the Southern Ocean is very interesting. However, I would suggest the authors calculate the errors for both the slope and intercept for the North Atlantic data, then compare them with those from the Southern Ocean. Without showing the relevant errors, this comparison is meaningless. Also, what is the correlation coefficient of the North Atlantic data? Is the correlation actually significant (data are very scattered)?

(6) Some of the discussion (e.g., Sections 4.1.1, 4.4) and conclusions in this manuscript depend heavily on unpublished data (e.g., Lemaitre et al., in prep; Roukaerts et al., unpublished data). These data are not accessible to readers and reviewers, and thus the discussion and conclusions reached cannot be justified. The authors need to either add these unpublished data to the manuscript, or remove relevant discussion.

Minor comments: L54-58: out of place. Move down to L66. L57-58: Inappropriate citations of Cao et al., 2016 and Horner et al., 2015. Both of these studies did not measure Baxs, but water column Ba isotopes. L87-91: It would be helpful to draw the subarctic front and formation site of the Labrador Sea Water in either Fig 2. L93-96: Are these results from this study or others'? L101-102: This sentence is confusing. Table S1 does not show PP or POC fluxes. L127-128: Were filters rinsed with MQ and dried at sea as well? Or were they kept frozen until home analysis? L129: 'for 4 h', not 'during' L142: 'at similar depths' L142-143: the sentence reads as "the comparison . . . was excellent". Please rewrite this sentence. L147-149: It would help to explain why only these few discrete samples were scanned. L178-186: This belongs to Methods. L187-188: The FE-SEM result is two fold of that measured by ICP-MS! L200-201: Some of these stations are not listed on Table S1 or shown on map. L204: These maxima appear to be at 200 – 600 m in Fig 4, not 100 – 300 m. It also doesn't seem that such maxima necessarily spread over a larger depth range. L210: Reference for

the Th-234 data. L210-211: Not clear why this needs to be mentioned here. L214: It needs to be justified why such value (180 pmol/L) is chosen as the background value. L218: The Baxs value is different from that in L200 L221-222: There is no double peak at St. 32: the Baxs values between 200 and 450 m are the same within errors. L225: To be scientifically correct: [Baxs] reach $\sim$ 750 pmol/L at 200-400 m. L226-227: This sentence is confusing. L228-232: Do vertical profiles between GEOSECS and GEOVIDE stations agree with each other, or do only the ranges agree? Since the ranges in [Baxs] are quite large, comparison of these ranges is meaningless unless you can show the comparison in a plot. L 241-244: Since there is no difference between the 100-500 m and 100-1000 m depth intervals, it is unclear to me why the 100-1000 m interval represent "the best the complete mesopelagic layer". L 245-146: delete "between 100 and 1000 m", as it is already specified that this is the mesopelagic depth interval used. L250-251: move to L244 L297-301: It is unclear how the advected signal was calculated. L297-298: Repetition of L221-222. L302-304: Please also speculate what causes the second Baxs peak at St. 38. L341-343: Repetition of L155-157. L355-359: These fit better in Section 4.3. L364-366: 100 – 1000 m, to be consistent. L369: '. . .is in the same order of magnitude as to. . .' L374-377: This sentence is difficult to read. L378: what does it mean '. . . with the region around Cape Verde. . .'? L381: similar to, not similar than

Fig 4 is referenced after Fig 5 in the text. Fig 5: corresponding to Section 4.1.2, this figure would benefit if depth ranges of major water masses are superimposed. Fig 6: Make the color coding consistent with Fig 1. Fig S1 is never referenced in the main text.

---

## Author Comment (AC1) · 25 Jan 2018

*General comments: This manuscript was a real pleasure to read both with respect to science presented and the way the manuscript is written and structured. I recommend the publication of this manuscript as it is with optional minor corrections.*

We would like to thank the referee for these very positive comments. All suggestions have been taken into account and are detailed below.

- *Lines 87-92: This sentence is too long and therefore confusing. I suggest splitting it into two shorter sentences.*

We agree and this has been changed in the revised manuscript.

Lines 99-104: Finally, as described elsewhere (Daniault et al., 2016; García-Ibáñez et al., 2015; Kieke and Yashayaev, 2015; Zunino et al., 2017; this issue), these provinces also differ in terms of their hydrographic features. The NADR province is crossed by the sub-arctic front (SAF), which was located near Station 26 during GEOVIDE (Fig. 2). Strong currents were observed near the Greenland margin (probably influencing Stations 51 and 64), and an intense 1500 m-deep convection happened during the winter preceding GEOVIDE in the central Labrador Sea (Station 69) due to the formation of the Labrador Sea Water (LSW) in winter (Fig. 2).

- *Line 93 and 95: Please state the depth of the upper ocean POC export flux*

OK.

Lines 105-108: The highest POC export fluxes from the upper-ocean (calculated at the depth "z" ranging from 30 to 110 m at Station 44 and 32, respectively) were observed in the NADR province and in the Labrador Sea and reached up to 10 mmol C $m^{-2}$ $d^{-1}$ at Station 69 (Lemaitre et al., 2018; this issue).

- *Line 96: For the transfer efficiency, I suggest to replace 'e.g.' with 'here defined as'; again, please state the depth of upper ocean POC export flux.*

OK.

Lines 109-111: The transfer efficiency (defined as the ratio of the POC export at z+100 m over the POC export at z) was more variable, ranging from 30% at Station 69 to 85% at Station 26 (Lemaitre et al., 2018; this issue).

- *Lines 228-232: The comparison of the Baex inventory at stations 44 and 51 to the GEOSECS Baex concentration data would benefit from adding the latter to the respective individual profiles in Figure 5.*

We agree and Figure 5 has been changed:

[Figure]

Figure 1: Vertical profiles of Ba$_{xs}$ concentrations (in pmol L$^{-1}$) determined from Niskin casts during GEOVIDE (squares) and GEOSECS (circles) cruises. The vertical black dashed line (at 180 pmol L$^{-1}$) represents the deep-ocean Ba$_{xs}$ value (or Ba$_{xs}$ background signal; Dehairs et al., 1997). The approximate depth range of the major water masses is also indicated in blue shading.

- *Lines 334-336: What are the R$^2$ and p values for the new relationship between Ba and oxygen consumption with and without station 44? Both R$^2$ and p values should feature the regression in Figure 8.*

With Station 44: R$^2$ = 0.33 and p-value = 0.07
Without Station 44: R$^2$ = 0.63 and p-value = 0.006
The values, R$^2$ and p-value, without considering Station 44 have been added to Figure 8, and the values including Station 44 have been added to the figure caption.

[Figure]

**Figure 2:** Regression of DWA mesopelagic $Ba_{xs}$ (pmol $L^{-1}$) versus $O_2$ consumption rate (µmol $L^{-1}$ $d^{-1}$) using the Southern Ocean transfer function from Dehairs et al. (1997; red circles) and the transfer function obtained here for the North Atlantic (black circles). Station 44 (triangle) was excluded from the regression. If station 44 is included, $R^2$=0.33 and p-value= 0.07.

- *Lines 649: Table 3 shows a comparison of Baex inventory and related POC remineralisation fluxes in the global ocean. The manuscript will benefit from having MR fluxes plotted on a global map.*

OK. We generated a new figure, Figure 9:

Figure 9:

[Figure]

POC remineralization fluxes (mmol m⁻² d⁻¹)

Figure 3: Summary of published POC remineralisation fluxes (in mmol C m⁻² d⁻¹) in the World's Ocean. The remineralisation fluxes for the Pacific Ocean (Dehairs et al., 2008) and the Southern Ocean (Cardinal et al., 2005; Jacquet et al., 2008a, 2008b, 2011b, 2015; Planchon et al., 2013) were calculated based on the $Ba_{xs}$ inventories. Insert shows data for the North Atlantic: sites indicated by circles lined in black are from the present study; at sites labelled with # symbols remineralisation was deduced from POC fluxes recorded by moored sediment traps (Honjo et al., 2008); at sites labelled by ° remineralisation was obtained from on-board incubations (Collins et al., 2015; Giering et al., 2014); sites labelled with * are GEOSECS sites for which we calculated remineralisation from existing $Ba_{xs}$ profiles (Brewer et al., unpublished results). Data were plotted using the ODV software (Schlitzer, 2017).

- *Line 705: The legend of Figure 5 should include the reference to the dashed line.*

OK. The legend was modified as follows:

Figure 4: Vertical profiles of $Ba_{xs}$ concentrations (in pmol L⁻¹) determined from Niskin casts during GEOVIDE (squares) and GEOSECS (circles) cruises. The vertical black dashed line (at 180 pmol L⁻¹) represents the deep-ocean $Ba_{xs}$ value (or $Ba_{xs}$ background signal; Dehairs et al., 1997). The approximate depth range of the major water masses is also indicated in blue shading.

- *Lines 735-738: The legend of Figure 9 should acknowledge the use of Ocean Data View (Schlitzer et al. 2004).*

Right, thank you for notifying this. Please, see above the new legend.

---

## Author Comment (AC2) · 25 Jan 2018

*General comments: Overall, the data presented are very interesting and contribute to further our knowledge on the biological carbon pump efficiency. However, the current manuscript can be significantly improved to avoid the redundancy throughout the text, to avoid heavy citation of data that are currently not available, and to make clear justification of some of their conclusions.*

We thank the referee for all the suggestions on how to improve our manuscript. We paid special attention to the introduction, limiting the redundancy and improving our conclusions. We hope that you will be satisfied with our detailed answers below.

- *Introduction: This section is unsatisfyingly short in my opinion. It really only includes 12 lines of background (L41 – 53). It needs to be expanded to include our current knowledge of the biological carbon pump (BCP) to date: why is it important to study BCP? Why North Atlantic? Why use the $Ba_{xs}$ proxy? What is the driving hypothesis and significance of your research?*

We agree and we have added more details to the Introduction, which now reads as follows:

Lines 42-67: The ocean represents the largest active $CO_2$ sink (Sabine et al., 2004) partly materialized by the oceanic biological carbon pump (BCP), which controls the export of carbon and nutrients to the deep ocean through the production of biogenic sinking particles (Boyd and Trull, 2007; Sigman and Boyle, 2000; Volk and Hoffert, 1985). The North Atlantic sustains one of the most productive spring phytoplankton bloom of the world's ocean (Esaias et al., 1986; Henson et al., 2009; Longhurst, 2010; Pommier et al., 2009). The high primary productivity in the North Atlantic in combination with the water mass formation there as part of the thermohaline circulation (Seager et al., 2002), results in a particularly efficient BCP in the North Atlantic (Buesseler et al., 1992; Buesseler and Boyd, 2009; Herndl and Reinthaler, 2013; Honjo and Manganini, 1993; Le Moigne et al., 2013b), estimated to contribute up to 18% of the global oceanic BCP (Sanders et al., 2014). However, the magnitude of the carbon transfer to the deep ocean depends on many factors including the efficiency of bacterial remineralisation within the mesopelagic layer (100–1000 m depth layer). In this layer, most of the particulate organic carbon (POC) exported from the upper mixed layer is respired or released to the dissolved phase as dissolved organic carbon (DOC; Buesseler et al., 2007; Buesseler and Boyd, 2009; Burd et al., 2016; Herndl and Reinthaler, 2013; Lampitt and Antia, 1997; Martin et al., 1987). Mesopelagic remineralisation has been often reported to balance or even exceed the carbon supply from the surface (i.e., POC and DOC; Aristegui et al., 2009; Baltar et al., 2009; Burd et al., 2010; Collins et al., 2015; Fernández-castro et al., 2016; Giering et al., 2014; Reinthaler et al., 2006), highlighting the impact of mesopelagic processes on bathypelagic carbon sequestration. Unfortunately, studies focusing on the mesopelagic layer are scarce, and the remineralisation process in this part of the water column remains poorly constrained. A variety of methods have been used to assess deep remineralisation. The attenuation of the particulate organic matter concentration with depth can be deduced from POC fluxes recorded by bottom tethered or free-floating neutrally buoyant sediment traps (e.g., Buesseler et al., 2007; Honjo et al., 2008; Martin et al., 1987) deployed at different depths. Bacterial respiration can be assessed by measuring the rate of dissolved oxygen consumption, but this approach is usually limited to the upper 200 m of depth because of sensitivity issues (Arístegui et al., 2005; Christaki et al., 2014; Lefèvre et al., 2008). However, sediment traps and direct respiration measurements are insufficiently reliable for depths exceeding 200 to 500 m (i.e. the lower

mesopelagic area). Earlier work has revealed that the accumulation of particulate biogenic barium (excess barium; $Ba_{xs}$) in the mesopelagic water column (100 – 1000 m) is related with organic carbon remineralisation.

- *Sampling and analyses: It is unclear why two completely different sampling and digesting methods were carried out for $Ba_{xs}$. While the authors gave great details on how different the methods were, there is little discussion on the comparison except in L207-210. The authors suggested good agreement between both datasets (their Fig S1); however, a closer look at the data only show good agreement for $Ba_{xs}$ < 400 pmol/L. At $Ba_{xs}$ > 400 pmol/L, data in Fig S1 (Go-Flo $Ba_{x}s$ vs. Niskin $Ba_{xs}$) are more scattered. These are actually the samples that are from depths of interests (100 – 1000 m), and likely suggest a discrepancy between the Go-Flo samples and Niskin ones. My other curiosity is, despite two different sampling techniques, why the authors applied two completely different filter digestions. Wouldn't it be better to use the same chemical protocols for a better data comparison?*

We understand your concern. The objectives of both sampling techniques were different. The main goal of the Niskin sampling was to derive $Ba_{xs}$ concentrations and, thus, carbon remineralisation fluxes in the mesopelagic zone (high resolution in the 100-1000m layer) at stations where primary productivities and carbon export fluxes were also determined. The goal of the Go-Flo sampling was, at first, dedicated to the determination of all dissolved and particulate trace elements and their isotopes. Since particulate Ba and Al were determined on samples collected by the Go-Flo bottles, we took the opportunity to compare both datasets in order to assess the quality of our data. We propose to add the following sentence to clarify this choice:

Lines 113-118: For different objectives, during GEOVIDE, suspended particles were collected by different sampling techniques. The main goal of the Niskin sampling was to derive $Ba_{xs}$ concentrations and, thus, carbon remineralisation fluxes in the mesopelagic zone (high resolution in the 100-1000 m layer) at stations where PP data and carbon export fluxes were also determined. The goal of the Go-Flo sampling was, at first, dedicated to the determination of all dissolved and particulate trace elements and their isotopes. Since particulate Ba and Al were determined on samples collected by both sampling techniques, we took the opportunity to compare both datasets in order to assess the quality of our data.

In the following graph, we plotted the $Ba_{xs}$ concentrations determined between 100 and 1000 m for the Niskin and Go-Flo systems. The regression is still significant ($R^2$=0.61 and p<0.01) and the slight discrepancy for the concentrations > 400 pM does not concern only samples from the depths of interest. We thus propose to modify the text and Figure S1 by the following:

Lines 162-164: For stations where total pBa and pAl concentrations were available at similar depths, the regression of $Ba_{xs}$ concentrations (100-1000 m layer) from the Go-Flo samples vs. those of the Niskin samples was significant (regression slope: 0.87; $R^2$: 0.61; p<0.01; n=66; Fig. S1) despite some discrepancies, especially in the higher concentration domain.

Lines 205-206: (i) both data sets converge (regression slope: 0.87; $R^2$: 0.61; p<0.01; n=66).

[Figure]

**Figure S1:** Comparison between the $Ba_{xs}$ concentrations obtained on samples collected by two different sampling and analytical methods (Niskin, 0.40 µm polycarbonate filters; and Go-Flo systems, paired 0.45 µm polyethersulfone and 5 µm mixed ester cellulose filters, see Gourain et al., 2018; this issue) for the 100-1000 m layer. See text for details.

Regarding the different chemical protocols, the referee is completely right: the fact to apply the same chemical protocol would have been better for comparing both datasets. Especially because the different HF concentrations between both protocols might be the reason of the slight discrepancy between both datasets. Moreover, the filter types were different (0.4 µm polycarbonate filters with the Niskin and paired 0.45 µm polyestersulfone / 5 µm mixed ester cellulose filters for the Go-Flo bottles), which can also lead to significant differences (Planquette and Sherrell, 2012). Nonetheless, despite different digestion methods and filters, we are satisfied by the comparison of both datasets. In order to discuss a little bit more on the differences between both datasets, we propose to add these sentences:

Lines 165-172: Such discrepancies could have resulted from differences in the chemical protocols and most likely the filters used. The Niskin samples collected on 0.4 µm polycarbonate filters were digested using a tri-acid mix (50% HCl/33% HNO$_3$/17% HF), while the Go-Flo samples collected on paired 0.45 µm polyestersulfone / 5 µm mixed ester cellulose filters were digested using a 50% HNO$_3$/10% HF acid mix. The use of different filter types has been shown to lead to different concentrations, depending on the element of consideration, despite using the same digestion technique (Planquette and Sherrell, 2012). The addition of HCl has been shown to not improve elemental recoveries of marine particles (Ohnemus and Lam, 2014) but the larger HF concentration of the tri-acid mix used for digesting the Niskin samples, likely, dissolved more of the refractory particles, explaining the slightly higher concentrations obtained for of these samples.

- *Section 3.1: This section involves large amount of discussion and I suggest moving most of this section to Section 4. I have a couple thoughts about this section. Firstly, Fig. 3 does not show any biogenic fragments, and thus I cannot judge whether barite crystals were actually observed adjacent to biogenic fragments. Secondly, barite crystals are believed to form in the microenvironment of decaying organic matter, so it is probably expected that no barite crystal is seen in surface samples.*

We agree with your suggestion and this section has been splitted between the Methods (Lines 178-189) and the Discussion (Lines 277-293) parts.

We also agree on the fact that quality of Fig. 3 was not acceptable as it was. We thus decided to zoom out or chose other images in order to see more clearly the biogenic fragments surrounding the barite crystals.

[Figure]

Figure 1: Barite particles observed by FE-SEM at (a) Station 38 (300 m); (b and c) Station 44 (700 m); (d) Station 69 (600 m). (c) is the backscattered electron image of the aggregate in (b) highlighting the shape of the partly hidden barite crystal. White arrows indicate the position of barite crystals.

The referee is right on the fact that barite crystals are not expected in surface samples, as he/she stated, barite crystals are formed in the microenvironment of decaying organic matter. To clarify this point, we added information in the section 4.1.

Lines 262-265: This result was expected as it fits in the concept of barite formation proposed by Stroobants et al. (1991), showing that the barium sulphate in biogenic aggregates of surface waters is not crystallized, whereas below this surface layer, when organic matter degradation occurs, barite is present as discrete micron-sized particles.

- *Section 3.3: This section is difficult to read, as there is a lot of jumping back and forth between provinces. I would suggest the authors to describe their data in a consistent way. For the selection of background level depth, the authors need to justify whether the absolute background values are more important for data comparison, or whether it is better to compare data consistently at the same depth.*

We made the following changes to improve the clarity of the section:
We first discussed about the 100-500 m and 100-1000 m depth interval of the DWA $Ba_{xs}$ content, and then about the background level depth. The integration over the 100-1000 m layer was used for comparing the different stations in a consistent way. Indeed, there are not significant changes between the DWA $Ba_{xs}$ contents integrated between the 100-1000 m depth layer and between 100 m and the real background level depth. Finally, the DWA $Ba_{xs}$ contents were described per province.

Lines 240-255: The $Ba_{xs}$ concentrations were integrated (trapezoidal integration) over two depth intervals of the mesopelagic layer (100–500 m and 100–1000 m; Table 2) to obtain depth-weighted average (DWA) $Ba_{xs}$ values.
The DWA $Ba_{xs}$ values between 100 and 500 m ranged from 399 to 672 pmol $L^{-1}$ and from 315 to 727 pmol $L^{-1}$ between 100 and 1000 m (Stations 51 and 69, respectively). The DWA $Ba_{xs}$ values varied by less than a factor of 1.4 between both modes of integration. Only for the Labrador Sea (Stations 64, 69 and 77) the DWA $Ba_{xs}$ values for the 100–1000 m were larger than for the 100-500 m interval. For the latter stations, the $Ba_{xs}$ inventories for the interval between 100 m and the depths were concentrations decreased to background level (1300, 1700 and 1200 m for Go-Flo casts at Stations 64, 69 and 77, respectively) were somewhat smaller than for the inventories between 100–1000 m (up to

1.5 times in the case of Station 77). To facilitate inter-comparison between stations, we consistently considered $Ba_{xs}$ inventories over the 100-1000 m depth interval in the following discussion.

Within the NAST province, Station 13 was characterized by a relatively low DWA $Ba_{xs}$ value of 419 pmol $L^{-1}$. Similarly, low median DWA $Ba_{xs}$ contents were observed within the NADR province (403 ± 34 pmol $L^{-1}$, n=4), with the lowest DWA $Ba_{xs}$ observed at Station 26 (391 ± 58 pmol $L^{-1}$).

The highest median DWA $Ba_{xs}$ value was observed in the ARCT province (566 ± 155 pmol $L^{-1}$, n=5). There, the DWA $Ba_{xs}$ contents were more variable between stations, ranging from 315 pmol $L^{-1}$ at Station 51 to 727 pmol $L^{-1}$ at Station 69, with a high DWA $Ba_{xs}$ also observed at Station 44 (633 pmol $L^{-1}$).

- *Section 4.2: The relationship of Baxs and carbon remineralization and their derived correlation in the North Atlantic are shown here, and the comparison between these and those in the Southern Ocean is very interesting. However, I would suggest the authors calculate the errors for both the slope and intercept for the North Atlantic data, then compare them with those from the Southern Ocean. Without showing the relevant errors, this comparison is meaningless. Also, what is the correlation coefficient of the North Atlantic data? Is the correlation actually significant (data are very scattered)?*

Section 4.3 (previously 4.2) has been modified. First, we moved Section 2.3 into Section 4.3 for more clarity.

Then, we showed that the new regression for the North Atlantic is significant ($R^2$=0.63 and p-value=0.006). Both $R^2$ and p-value have been added to Figure 8.

We also calculated and indicated in the text and in Fig. 8 the errors for both the slope and intercept for the North Atlantic data:

Slope: 23,391 ± 6,368

Intercept: 247 ± 61

Indeed, when taking into account these errors, the new North Atlantic regression is not significantly different from the one of the Southern Ocean. In consequences, we modified the section as follow:

Lines 343-386: In previous studies focusing on the Southern Ocean, $Ba_{xs}$ based-mesopelagic carbon remineralisation fluxes were estimated using Eq. (2), which relates the accumulated mesopelagic $Ba_{xs}$ inventory to the rate of oxygen consumption (Shopova et al., 1995; Dehairs et al., 1997):

$$\text{Mesopelagic } Ba_{xs} = 17200 \times JO_2 + Ba_{residual} \quad\quad (2)$$

where M*esopelagic Ba_{xs}* is the depth-weighted average in the mesopelagic layer (DWA; in pmol $L^{-1}$), $JO_2$ is the rate of oxygen consumption (in µmol $L^{-1}$ $d^{-1}$), and $Ba_{residual}$ is the deep-ocean $Ba_{xs}$ value observed at zero oxygen consumption (or $Ba_{xs}$ background signal), which was determined to reach 180 pmol $L^{-1}$ (Dehairs et al., 1997).

The oxygen consumption $JO_2$ can be converted into a C remineralisation flux through Eq. (3):

$$\text{POC mesopelagic remineralisation} = Z \times JO_2 \times (C:O_2)_{Redfield\ Ratio} \quad\quad (3)$$

where the *POC mesopelagic remineralisation* is in mmol C $m^{-2}$ $d^{-1}$, *Z* is the thickness of the layer in which the mesopelagic $Ba_{xs}$ is calculated, $JO_2$ is the rate of oxygen consumption given by Eq. (2) and *(C:O_2)_{Redfield Ratio}* is the stoichiometric molar ratio of carbon to dioxygen (127/175; Broecker et al., 1985). However, it is of interest to investigate if this relationship can be applied in the North Atlantic. Therefore, we determined the oxygen utilization rate (OUR; µmol $kg^{-1}$ $yr^{-1}$), which is obtained by dividing the apparent oxygen utilization (AOU, in µmol $kg^{-1}$) by the water mass age (Table S1). From the Iberian coast to Greenland, the age calculation was based on the CFC-12 distribution (when

available, otherwise CFC-11) determined in 2012 (OVIDE CARINA cruise, de la Paz et al., 2017). For the Labrador Sea, the mean age of LSW has been estimated by Rhein et al. (2015) based on a 25 year record of CFC contents. The OUR was then integrated over the 100–1000 m layer. The resulting regression between DWA $Ba_{xs}$ and OUR is as follows (see Fig. 8):

$$Mesopelagic\ Ba_{xs} = 23391\ (\pm6368) \times JO_2 + 247\ (\pm61) \qquad (4)$$

where M*esopelagic Ba$_{xs}$* and *JO$_2$* are defined in Eq. 2. Here, *Ba$_{residual}$* is 247 pmol L$^{-1}$.

This regression is significant ($R^2$ = 0.63; p-value = 0.006) when Station 44 is excluded. This latter station was located in the Irminger Gyre (Zunino et al., 2017; this issue) and it is possible that the gyre system induced an accumulation and retention of mesopelagic $Ba_{xs}$, which then no longer reflects remineralisation associated with the present growth season.

Figure 8 also shows the oxygen consumption related to the GEOVIDE $Ba_{xs}$ values using the Southern Ocean regression (Eq. 2). It appears that for a given mesopelagic $Ba_{xs}$ inventory the oxygen consumption is smaller when using the Southern Ocean regression. However, both regressions are not significantly different when taking into account the errors associated with the slope and intercept of the regression in Eq. 4. The Southern Ocean regression appears to represent a lower limit that seems to over-estimate the remineralisation fluxes. Furthermore, the relationship here deduced for the North Atlantic is sensitive to potential errors. Indeed, calculation of OUR has been shown to under-estimate the ocean respiration because of the non-proportional diffusive mixing of AOU and water mass age resulting in an excess loss of AOU versus age (Koeve and Kähler, 2016). This would decrease the mismatch between the Southern Ocean and North Atlantic regressions. Errors can also be directly associated with the CFC-based age values of the water masses, which would appear especially critical for LSW. Indeed, the severe winter preceding the cruise (2013/2014) appeared to have strongly ventilated LSW with a mixed layer depth exceeding 1700 m (Kieke and Yashayaev, 2015), indicating that the mean age (4 years) estimated by Rhein et al. (2015) may have over-estimated the real LSW age (P. Lherminier, personal communication). Moreover, in the Labrador Sea, the residence time of LSW strongly varies between the central Labrador Sea (4–5 years) and the boundary currents off the Greenland and Newfoundland coasts (a few months; Deshayes et al., 2007; Straneo et al., 2003). An over-estimation of these ages leads to under-estimating OUR, resulting in reducing the apparent discrepancy between the both North Atlantic and Southern Ocean regressions.

In the following discussion, carbon remineralisation fluxes are estimated for the North Atlantic (GEOVIDE and GEOSECS cruises) using Eq. (4) and (3).

- *Some of the discussion (e.g., Sections 4.1.1, 4.4) and conclusions in this manuscript depend heavily on unpublished data (e.g., Lemaitre et al., in prep; Roukaerts et al., unpublished data). These data are not accessible to readers and reviewers, and thus the discussion and conclusions reached cannot be justified. The authors need to either add these unpublished data to the manuscript, or remove relevant discussion.*

The primary production and POC export data at each station, even if not published yet, are indicated in the Table 4. The POC export fluxes will be submitted in a paper of this special issue, referenced as "Lemaitre et al., 2018; this issue". The PP data will be published in two different papers "Fonseca-Batista, 2018" and "Lemaitre et al., 2018", which will be part of this issue as well.

- *L54-58: out of place. Move down to L66.*

Done.

- *L57-58: Inappropriate citations of Cao et al., 2016 and Horner et al., 2015. Both of these studies did not measure Baxs, but water column Ba isotopes.*

Both references have been removed.

- *L87-91: It would be helpful to draw the subarctic front and formation site of the Labrador Sea Water in either Fig 2.*

Figure 2 has been modified accordingly.

- *L93-96: Are these results from this study or others'?*

The POC export fluxes, export and transfer efficiencies during the GEOVIDE cruise will be published in "Lemaitre et al., 2018", a paper part of this issue.

- *L101-102: This sentence is confusing.*

We propose to rewrite the sentence as follows:

Lines 120-121: At eleven station, 18 depths were generally sampled between the surface and 1500 m in order to cover a high vertical resolution in the mesopelagic layer (Table S1).

- *Table S1 does not show PP or POC fluxes.*

PP and POC export fluxes are indicated in Table 4.

- *L127-128: Were filters rinsed with MQ and dried at sea as well? Or were they kept frozen until home analysis?*

A sentence has been added:

Lines 146-147: Excess seawater from the filters was drawn off and then the filters were frozen in acid-cleaned petri-dishes until home analysis.

- *L129: 'for 4 h', not 'during'*

This has been changed.

- *L142: 'at similar depths'*

OK.

- *L142-143: the sentence reads as "the comparison…was excellent". Please rewrite this sentence.*

Done.

Lines 162-164: For stations where total pBa and pAl concentrations were available at similar depths, the regression of $Ba_{xs}$ concentrations (100-1000 m layer) from the Go-Flo samples vs. those of the Niskin samples was significant (regression slope: 0.87; $R^2$: 0.61; $p<0.01$; n=66; Fig. S1) despite some discrepancies, especially in the higher concentration domain.

- *L147-149: It would help to explain why only these few discrete samples were scanned.*

We propose to change the sentence as:

Lines 174-177: Because of time consuming analyses, seven filters from the different basins were scanned: six samples with high mesopelagic $Ba_{xs}$ concentrations (Station 13 at 400 m; Station 38 at 300 m; Station 44 at 300 and 700 m; Station 69 at 600 m and Station 77 at 300 m) and one sample with high surface $Ba_{xs}$ concentrations (Station 26 at 50 m). For each sample, a filter surface of 0.5 cm² was analysed.

- *L178-186: This belongs to Methods.*

OK. This section has been moved to the methods section.

- *L187-188: The FE-SEM result is two folds of that measured by ICP-MS!*

It is true that there is a difference between both technics (of a factor 1.5). However, one must take into account that i) the very small fraction of filter analysed may not be truly representative of the whole filter, leading to a large error and ii) barite crystals were assimilated as ellipses, which can also lead to some error. We therefore find the concentration obtained by the FE-SEM method remarkably similar to the one obtained by ICP-MS.

- *L200-201: Some of these stations are not listed on Table S1 or shown on map.*

All stations (sampled by Go-Flo and Niskin bottles) are shown in Figure 2 and 4. Only the stations sampled by Niskin bottles are listed in Table S1 because they are the ones used for calculating the remineralisation fluxes.

- *L204: These maxima appear to be at 200 – 600 m in Fig 4, not 100 – 300 m. It also doesn't seem that such maxima necessarily spread over a larger depth range.*

OK. The sentence has been changed to:

Lines 202-203: These maxima occurred between 200 and 600 m but were spread over larger depth intervals in the ARCT province, where high $Ba_{xs}$ values occurred until 1200 m depth at Station 69.

- *L210: Reference for the Th-234 data.*

OK. Now included.

- *L210-211: Not clear why this needs to be mentioned here.*

The sentence has been removed.

- *L214: It needs to be justified why such value (180 pmol/L) is chosen as the background value.*

Right. We added these sentences:

Lines 209-213: All the vertical $Ba_{xs}$ profiles (Fig. 5) show increased concentrations between 100 and 1000 m, followed by lower concentrations deeper that tend to return to a background level of 180 ± 54 pmol $L^{-1}$ (n=10) as average along the GEOVIDE transect. This background value is quite characteristic for the deep ocean (> 1000 m) and is considered to represent the residual $Ba_{xs}$ left over after partial

dissolution and sedimentation of Ba$_{xs}$ produced during previous phytoplankton growth events (Dehairs et al., 1997).

- *L218: The Baxs value is different from that in L200*

This is because the concentration given in Section 3.1.1 has been determined at 35 m via the Go-Flo system, while the concentrations given in Section 3.1.2 has been determined at 50 m via the Niskin system. In section 3.1.2, we focused on samples taken with the Niskin bottles.

For more clarity, we added this sentence in the text:

Line 219-220: (note that the value for the Go-Flo sample at 35 m reaches 24643 pmol L$^{-1}$; section 3.1.1).

- *L221-222: There is no double peak at St. 32: the Baxs values between 200 and 450 m are the same within errors.*

Right, we changed the sentence accordingly.

- *L225: To be scientifically correct: [Baxs] reach _ 750 pmol/L at 200-400 m.*

OK. The sentence has been modified:

Lines 225-226: In the ARCT province, a similar double peak profile was observed at Station 44, in the Irminger Sea, with Ba$_{xs}$ concentrations reaching 750 pmol L$^{-1}$ between 200 and 400 m and 820 pmol L$^{-1}$ at 700 m.

- *L226-227: This sentence is confusing.*

OK, we changed it as:

Lines 227-229: Close to the Greenland margin (Station 51), Ba$_{xs}$ concentrations reached a maximum of 495 pmol L$^{-1}$ at 300 m, which was lower than the maxima determined at the other stations of the ARCT province.

- *L228-232: Do vertical profiles between GEOSECS and GEOVIDE stations agree with each other, or do only the ranges agree? Since the ranges in [Baxs] are quite large, comparison of these ranges is meaningless unless you can show the comparison in a plot.*

The GEOSECS profiles have been added to Fig. 5. Only the ranges agree between profiles.

- *L 241-244: Since there is no difference between the 100-500 m and 100-1000 m depth intervals, it is unclear to me why the 100-1000 m interval represent "the best the complete mesopelagic layer".*

In this light, the paragraph has been modified as follows:

Lines 243-249: The DWA Ba$_{xs}$ values varied by less than a factor of 1.4 between both modes of integration. Only for the Labrador Sea (Stations 64, 69 and 77) the DWA Ba$_{xs}$ values for the 100–1000 m were larger than for the 100-500 m interval. For the latter stations, the Ba$_{xs}$ inventories for the interval between 100 m and the depths were concentrations decreased to background level (1300, 1700 and 1200 m for Go-Flo casts at Stations 64, 69 and 77, respectively) were somewhat smaller than for the inventories between 100–1000 m (up to 1.5 times in the case of Station 77). To facilitate intercomparison between stations, we consistently considered $Ba_{xs}$ inventories over the 100-1000 m depth interval in the following discussion.

- *L 245-146: delete "between 100 and 1000 m", as it is already specified that this is the mesopelagic depth interval used.*

Done.

- *L250-251: move to L244*

OK. This section has been modified. See our answer to your major comment number 4.

- *L297-301: It is unclear how the advected signal was calculated.*

Right. We added this sentence:

Line 314-315: We calculated the DWA $Ba_{xs}$ without taking into account the 2$^{nd}$ peak (100-600 m depth interval) subtracted it from the total DWA $Ba_{xs}$ (100-1000 m depth interval) to estimate the advected signal.

- *L297-298: Repetition of L221-222.*

OK. We deleted the repetitive parts.

- *L302-304: Please also speculate what causes the second Baxs peak at St. 38.*

OK.

Lines 320-323: Association of $Ba_{xs}$ maxima with water masses is not always clear, as it is evident from the case of Station 38 where the second $Ba_{xs}$ maximum at 700 m (Fig. 5) does not coincide with a specific water mass (Fig. 7). In this case, the deep $Ba_{xs}$ maximum may possibly result from remineralisation generated by larger or heavier organic aggregates reaching greater depths.

- *L341-343: Repetition of L155-157.*

OK. To avoid this repetition, we moved the previous section 2.3 to the section 4.3.
Please, see the answer to your major comment number 5, or lines 344-387 in the revised manuscript.

- *L355-359: These fit better in Section 4.3.*

Right, a new sub-section has been added in Section 4.4.

Lines 388-395: 4.4.1. Remineralisation from the $Ba_{xs}$ proxy
The GEOVIDE remineralisation fluxes are compared with values reported for the World Ocean and also based on $Ba_{xs}$ inventories (Table 3; Fig. 9). In the North Atlantic, the fluxes obtained during the GEOVIDE and GEOSECS (symbolized by stars in Fig. 9) cruises are of the same order of magnitude, highlighting a relatively constant remineralisation over the last 44 years. The remineralisation fluxes reported for the Southern and Pacific Oceans are similar to those in the NAST and NADR provinces of the North Atlantic. However, the fluxes in the ARCT province are clearly higher, highlighting an important remineralisation in the northern part of the North Atlantic compared to other oceans.

- *L364-366: 100 – 1000 m, to be consistent.*

OK. Done.

- *L369: '…is in the same order of magnitude as to…'*

OK. Done.

- *L374-377: This sentence is difficult to read.*

OK. We splitted this sentence into two for more clarity.

Lines 410-413: Then, by difference, the authors estimated an annual average of carbon remineralisation fluxes in the mesopelagic layer, which were converted into daily average fluxes. Remineralisation fluxes reached values of 34 mmol C $m^{-2}$ $d^{-1}$ in the ARCT province, 9 mmol C $m^{-2}$ $d^{-1}$ in the NADR province and 4 mmol C $m^{-2}$ $d^{-1}$ in the NAST province (Fig. 9).

- *L378: what does it mean '…with the region around Cape Verde…'?*

For more clarity, we removed these details and added "the flux in the ARCT province was one of the highest" (line 413).

- *L381: similar to, not similar than*

OK.
- *Fig 4 is referenced after Fig 5 in the text.*

Right. In line 181, we now refer to Section 3 instead of Fig. 5 to illustrate the high mesopelagic $Ba_{xs}$ content at Station 69.

- *Fig 5: corresponding to Section 4.1.2, this figure would benefit if depth ranges of major water masses are superimposed.*

It is an interesting idea, thank you. The depth ranges of the major water masses have been added for each station.

Figure 5:

[Figure]

**Figure 2:** Vertical profiles of $Ba_{xs}$ concentrations (in pmol $L^{-1}$) determined from Niskin casts during GEOVIDE (squares) and GEOSECS (circles) cruises. The vertical black dashed line (at 180 pmol $L^{-1}$) represents the deep-ocean $Ba_{xs}$ value (or $Ba_{xs}$ background signal; Dehairs et al., 1997). The approximate depth range of the major water masses is also indicated in blue shading.

- *Fig 6: Make the colour coding consistent with Fig 1.*

OK. Here are the two new Figures:

Figure 1:

[Figure]

Figure 3: Satellite derived Chlorophyll-*a* concentrations (MODIS Aqua from http://giovanni.sci.gsfc.nasa.gov/), in mg m$^{-3}$ during the GEOVIDE cruise (May and June 2014). The GEOVIDE transect (grey line) and the main crossed provinces are indicated. NAST: North Atlantic Subtropical gyre; NADR: North Atlantic Drift; ARCT: Atlantic Arctic. Coloured circles indicate stations sampled at the corresponding month.

Figure 6:

[Figure]

Figure 6: Map of time averaged Chlorophyll-*a* concentrations (in mg m⁻³) for the period from January to June 2014 (monthly 4 km MODIS Aqua model; http://giovanni.sci.gsfc.nasa.gov/).

- *Fig S1 is never referenced in the main text.*

Figure S1 is referenced line 164 (Section 2.2).